# A multi-diagnostic approach to cloud evaluation

Keith D. Williams* and Alejandro Bodas-Salcedo

Met Office, Exeter, UK

*Corresponding author address: Keith Williams, Met Office, FitzRoy Road, Exeter, EX1 3PB, UK.

Email: keith.williams@metoffice.gov.uk Tel: +44 (0)1392 886905 Fax: +44 (0)1392 885681

April 26, 2017

## Abstract

Most studies evaluating cloud in general circulation models present new diagnostic techniques or observational datasets, or apply a limited set of existing diagnostics to a number of models. In this study, we use a range of diagnostic techniques and observational datasets to provide a thorough evaluation of cloud, such as might be carried out during a model development process. The methodology is illustrated by analysing two configurations of the Met Office Unified Model - the currently operational configuration at the time of undertaking the study (Global Atmosphere 6, GA6), and the configuration which will underpin the United Kingdom's Earth System Model for CMIP6 (Coupled Model Intercomparison Project 6) (GA7).

By undertaking a more comprehensive analysis which includes compositing techniques, comparing against a set of quite different observational instruments and evaluating the model across a range of timescales, the risks of drawing the wrong conclusions due to compensating model errors are minimised and a more accurate overall picture of model performance can be drawn.

Overall the two configurations analysed perform well, especially in terms of cloud amount. GA6 has excessive thin cirrus which is removed in GA7. The primary remaining errors in both configurations are the in-cloud albedos which are too high in most northern hemisphere cloud types and sub-tropical stratocumulus, whilst the stratocumulus on the cold air side of southern hemisphere cyclones has in-cloud albedos which are too low.

# 1    Introduction

The accurate simulation of cloud in general circulation models (GCMs) is of considerable importance across all timescales. At numerical weather prediction (NWP) timescales of a few days or less, cloud amount as a forecast product is of direct relevance to a number of users (e.g. aviation, solar farms, etc.) and affects forecasts of other variables through its radiative impact on the surface temperature and the effects of diabatic heating on the large scale circulation. On climate timescales, the radiative feedback from cloud on the global energy budget remains one of the largest uncertainties in determining the global climate sensitivity (Flato *et al*, 2013).

Traditionally, the evaluation of cloud has been limited to quantities which were perceived to be of interest to the end user such as ground-based observations of total cloud amount (Mittermaier, 2012), or top-of-atmosphere cloud radiative forcing (CRF) (e.g. Gleckler *et al*, 2008). However, compensating errors within GCMs can result in a model performing well on such a limited set of metrics, despite the processes within the model being in error. A classic example is the simulation of subtropical stratocumulus, for which many GCMs simulate too little cloud cover, but the cloud which is simulated is too bright, the two errors compensating to result in a reasonable CRF (e.g. Williams *et al*, 2003; Nam *et al*, 2012).

Over recent years, a range of process-orientated diagnostic techniques have been developed which composite the data according to other large-scale variables, with the intention of reducing the chances of a model appearing to perform well due to compensating errors. Compositing variables have, amongst others, included: large scale vertical velocity, (Bony *et al*, 2004); various measures of lower tropospheric stability, (Klein and Hartmann, 1993; Williams *et al*, 2006; Myers and Norris, 2015); position relative to cyclone centre, (Klein

and Jakob, 1999; Govekar *et al*, 2011) and cloud regime (Williams and Tselioudis, 2007).

In addition to model errors, there are errors in the observational datasets and how they are used for GCM evaluation. For example, the 'total cloud amount' obtained from ground-based ceilometers will be underestimated since they typically cannot detect the highest clouds. When these issues are known, they can be mitigated by sampling the model in a consistent manner to the observations (e.g. in this case, only considering model clouds up to the maximum height the ceilometer can detect). For cloud evaluation against satellite data, increasing use is being made of satellite simulators which aim to emulate the observations by carrying out a consistent retrieval on the model. A number of satellite simulators have been brought together in the CFMIP (Cloud Feedback Model Intercomparison Project) Observational Simulator Package (COSP; Bodas-Salcedo *et al*, 2011) which has now been included in many GCMs.

Arguably the best way to minimise issues around compensating model errors, observational error and model-observation comparison issues, is to routinely evaluate cloud in GCMs against a wide range of different observational datasets, using simulators where appropriate and using a range of diagnostic techniques in order to gain a consistent picture of model biases. In this study, we illustrate how the approach can be used for model development by applying a comprehensive cloud evaluation to two configurations of the Met Office Unified Model (UM).

Cloud errors in the UM, possibly more than any other variable, are very similar across timescales and horizontal resolutions (Williams and Brooks, 2008). Figure 1 shows the bias in high, mid and low cloud in the Global Atmosphere 6 (GA6; Walters *et al*, 2016) configuration of the UM against CALIPSO (Cloud-Aerosol Lidar and Infrared Pathfinder Satellite Observation). It can be seen that the day 1 and day 5 forecast biases at N320

resolution (40km in mid-latitudes) are very similar to each other and to a climatological

bias obtained from an AMIP (Atmosphere Model Intercomparison Project; Gates, 1992)

simulation at N96 resolution (135km in mid-latitudes). This means that we can make use

of each timescale in our analysis to its strengths and the conclusions should be applicable

across the systems. Although the UM is being used (a model which is routinely assessed

for both NWP and climate work), we consider the cross-timescale approach a key aspect of

the comprehensive evaluation. The initialised hindcasts provide case studies where model

biases can be investigated in detail for particular meteorological events, in situations

where the large scale dynamics remain close to those observed. In contrast, the longer

climate simulations provide characterisation and statistics of the systematic errors. For

those GCMs which are typically only used for a limited set of timescales, the AMIP

(Gates, 1992) and Transpose-AMIP (Williams *et al*, 2013) experimental designs allow the

possibility of this cross-timescale evaluation.

In the next section we provide details of the models, experiments and observational

data subsequently presented. We then evaluate the cloud simulation in the model over the

tropics, mid-latitude storm tracks and mid-latitude land in sections 3, 4 & 5 respectively.

The overall impact of the cloud on the global radiation balance is then discussed in

section 6. We summarise in Section 7.

# 2 Models and observational datasets

## a Models and experimental design

Two configurations of the UM are used in this study. GA6 has been operational in all

global model systems at the Met Office since 15th July 2014 and is fully documented by

Walters *et al* (2016). GA7 has recently been frozen and is documented by Walters *et al* (2017). It is intended that GA7 will form the physical atmosphere model used by the United Kingdom Earth System Model 1 (UKESM1) which will be submitted to CMIP6 (Coupled Model Intercomparison Project 6).

There are numerous physical parametrization changes between GA6 and GA7 which are detailed in Walters *et al* (2017). Those of most relevance for this study are:

1. The introduction of a scheme to allow the turbulent fluxes within the boundary layer capping inversion to be resolved and for clouds ('forced cumulus') to form within it. The height of the top of the capping inversion is diagnosed using an energetic argument based on Beare (2008) which is applied to the calculation of an ascending air parcel used for the diagnosing convection. Within the undulations of the capping inversion, if the parcel doesn't reach it's level of free convection then forced cumulus may form. The cloud-base cloud fraction is parametrized as varying linearly with cloud depth, between a minimum of 0.1 and a maximum of 0.3 for cloud depths between 100 m and 300 m, based loosely on the observations of (Zhang and Klein, 2013). Increments to the overall prognostic cloud fraction are calculated as necessary to increase it to the forced convective cloud fraction. The in-cloud water content is taken from the adiabatic parcel ascent in the cumulus diagnosis.

2. A package of changes designed to improve warm rain microphysics. This includes a change to the auto-conversion scheme to be based on Khairoutdinov and Kogan (2000) which was developed from a bin resolved microphysics scheme and so closely correspond to best estimates of what these process rates should be. They are upscaled to a GCM following Boutle *et al* (2014). Because microphysical process rates are nonlinear, calculating the process rate from in-cloud mean quantities

(as is done in GA6) can lead to large biases in the process rate in low resolution

GCMs where the sub-grid variability is significant. This parametrization corrects

the process rates for the presence of sub-grid variability, based on parametrizations

of the sub-grid variability derived from aircraft, CloudSat (Stephens *et al*, 2002)

and CloudNet-ARM (Atmosphere Radiation Measurement) site observations.

3. Improved cloud ice optical properties and ice particle size distributions (PSD) fol-

lowing Baran *et al* (2014) and Field *et al* (2007) respectively. The new PSD is an

empirical fit that is better supported by observations and in GA7 is used consistently

between the microphysics and radiation schemes.

4. Reduced rate of cirrus spreading by two orders of magnitude. The cirrus spreading

was a simple parametrization intended to account for the spreading of cirrus through

shear as it falls. It uses the model wind shear between successive layers to spread the

ice as it falls at a rate controlled by a tunable parameter. It was included, largely as

a tuning of outgoing longwave radiation (OLR), in an earlier configuration (GA4;

Walters *et al*, 2014) and it is desirable to reduce the effect until the scheme is

developed on firmer physical grounds.

5. Addition of the turbulent production of liquid water in mixed-phase clouds fol-

lowing Field *et al* (2014). An exactly soluble stochastic model is used to describe

sub-grid relative humidity fluctuations. The probability density function (PDF) of

the fluctuations in a model grid-box depends on the turbulent local state based

on the boundary layer turbulent kinetic energy and on any pre-existing ice cloud.

Increments to liquid water cloud prognostic fields are diagnosed from the PDF.

This increases the liquid water contents and volume fractions of liquid cloud. A

temperature threshold restricts the scheme to regions below 0 Celsius.

6. A change to the aerosol scheme from CLASSIC (Coupled Large-Scale Aerosol Simulator for Studies In Climate; (Bellouin *et al*, 2011)) to GLOMAP-mode (Global Model of Aerosol Processes modal aerosol scheme; (Mann *et al*, 2010)). GLOMAP-mode models the aerosol number, size distribution, composition and optical properties from a detailed, physically-based treatment of aerosol microphysics and chemistry. The scheme simulates speciated aerosol mass and number in 4 variable-size soluble modes to cover different aerosol size ranges (nucleation, aitken, accumulation and coarse modes) as well as an insoluble aitken mode. The prognostic aerosol species represented by GLOMAP-mode are sulphate, black carbon, organic carbon and sea salt. Cloud condensation nuclei are activated into cloud droplets using the Activate aerosol activation scheme based on Abdul-Razzak and Ghan (2000).

7. Although only small changes have been made to the scientific basis of the convection scheme, the numerics of the scheme have been re-written (the so called '6A convection scheme'). This is described in Walters *et al* (2017), but the key points are:

- Three iterations rather than one iteration is used to solve the implicit equations for the potential temperature of the detrained mass and the residual plume in the calculation of the forced detrainment.

- Three rather than two iterations are used in determining the potential temperature at saturation after lifting the the parcel from one level to the next under dry ascent. The evaporation of parcel condensate is now also allowed if the parcel becomes sub-saturated after entrainment and the dry ascent.

- The ascent in the 6A scheme will terminate when the mass flux falls below 5% of its value at cloud base, which replaces the previous arbitrary small value.

- The convection scheme will introduce small errors in the conservation of energy and water. These are now corrected locally to ensure that the column integral of these quantities is the same after the call to convection as they were before, replacing the previous global correction.

For each configuration, two types of experiment have been conducted, both being standard tests used within the model development cycle for proposed changes to the UM. These are a 20 year (1988-2007) AMIP experiment run at a horizontal resolution of N96 (135km in mid-latitude), and a set of 24 independent 5-day NWP hindcasts spread between December 2010 and August 2012, run at N320 (40km in mid-latitude) and initialised from European Centre for Medium range Weather Forecasts (ECMWF) analyses. ECMWF rather than Met Office analyses are used for case study tests within the model development cycle so as not to favour the performance of the control model which may have had the UM data assimilation system tuned towards it. This also makes the hindcasts consistent with the standard Transpose-AMIP experiment (Williams *et al*, 2013), except for the specific dates run.

## b  Observational datasets and simulators

We make use of a variety of observational datasets. The International Satellite Cloud Climatology Project (ISCCP) D1 product (Rossow and Schiffer, 1999) uses passive radiometer data from geostationary and polar orbiting satellites to produce 3-hourly histograms of cloud fraction on a $2.5^{o}$ grid in seven cloud top pressure and six optical depth bins. CALIOP (Cloud-Aerosol Lidar with Orthogonal Polarization) is a cloud lidar on the CALIPSO platform (Winker *et al*, 2010), which is part of the NASA A-train satellite constellation. It uses a nadir pointing instrument with a beam diameter of 70m at

the earth's surface and produces footprints every 333m in the along-track direction. We use the GCM-orientated CALIPSO cloud product (Chepfer *et al*, 2010) which contains histograms of cloud amount in joint height–backscatter ratio bins as well as total cloud amount in standard low (>680hPa), mid (440hPa–680hPa) and high (<440hPa) categories. The histograms are formed by assigning the cloud occurrence in each height and backscattering ratio category with a minimum backscattering ratio of 3. The percentage occurrence in each bin is then determined. CloudSat (Stephens *et al*, 2002), is a 94GHz cloud radar which pulses a sample volume of 480m in the vertical and a spatial resolution of 1.4km. We use the CloudSat 2B geometrical profile (2B-GEOPROF) (Marchand *et al*, 2008) product which includes histograms of hydrometeor frequency in joint height–radar reflectivity bins. The complementary nature of the CloudSat and CALIPSO in terms of the hydrometeor profile provided by the radar and detection of very thin clouds by the lidar, and their co-location on the A-train mean that they may be combined to produce a 'best estimate' hydrometeor fraction through the depth of the atmosphere column. This has been done by Mace and Zhang (2014) in the form of the radar-lidar geometrical profile (RL-GEOPROF) product. In this study we use revision 4 (R04) of RL-GEOPROF.

All of the above have a simulator within COSP (Bodas-Salcedo *et al*, 2011) in order to produce comparable diagnostics from the model by emulating the satellite retrieval. The simulators are described by Klein and Jakob (1999)/Webb *et al* (2001), Chepfer *et al* (2008), Haynes *et al* (2007) for the ISCCP, CALIPSO and CloudSat simulators respectively. The ISCCP simulator uses a perfect optical depth retrieval, taking into account the subgrid variability of cloud condensate used in the model's radiative transfer model. The cloud top pressure is based on a simple estimation of the 10.5 micron brightness temperature, which is then mapped onto the temperature profile as a function of pressure. The

CALIPSO and CloudSat simulators are forward models of the attenuated backscattering ratio at 532nm, and reflectivity at 94GHz, respectively.

COSP version 1.4 is used in this study, which does not include a diagnostic of combined radar-lidar cloud fraction. In order to compare model clouds against RL-GEOPROF, a new diagnostic that combines CALIPSO backscattering ratio and CloudSat reflectivities has been developed. The new diagnostic is a simple combined cloud mask. Each volume in each sub-column is flagged as cloudy if the CALIPSO backscattering ratio (SR) is above the detection threshold (SR≥3.0) or the CloudSat reflectivity is greater than -30dBZ. Then the cloud fraction at each level is calculated as the ratio of cloudy volumes divided by the total number of volumes.

The cloud identification of the GCM Orientated CALIPSO Cloud Product (GOCCP) is performed at the nominal horizontal resolution (330m below 8km, and 1km above 8km). At that resolution, the instrument noise level is high. In order to minimise false positives due to noise, GOCCP uses a very conservative backscattering ratio threshold (SR=5). The CALIPSO cloud mask used in the RL-GEOPROF product uses a 5km spatial averaging to increase the signal-to-noise ratio and allow the detection of thinner clouds. Chepfer *et al* (2013) show that the implicit SR detection threshold in the CALIPSO cloud mask used in RL-GEOPROF ranges between 1 and 3. We have therefore reduced the SR threshold from 5 to 3 in COSP in order to represent a diagnostic that is more comparable to the RL-GEOPROF cloud mask. A value of 3 is chosen because it is one of the boundaries used by GOCCP to construct height-SR histograms. Supplementary material Figure 1 shows the impact of reducing the SR threshold in the vertical profile of cloud fraction over the tropical belt.

Evaluation of the top-of-atmosphere radiative fluxes are made against CERES-EBAF

(Clouds and the Earth's Radiant Energy System–Energy Balanced and Filled) dataset (Loeb *et al*, 2009). We also make use of synoptic surface observation (SYNOP) data (WMO, 2008). Mittermaier (2012) discuss some of the issues around using these data for cloud verification. We consider the most significant for evaluation of model biases are the differences in the maximum altitude at which automated ceilometers used by different countries can detect cloud, which in turn differ from human observers. In this study we just use cloud base height information in situations where the cloud base is below 1km. It is in these situations that the SYNOP observations should be the most consistent and reliable.

Compositing techniques are employed to provide a more process-orientated cloud evaluation. In all cases, the data used to composite the observed cloud fields ($500hPa$ vertical velocity, pressure at mean sea level, etc.) are from ERA-I (ECMWF Interim Re-analyses; Dee *et al*, 2011). Composites using daily mean data are formed from 5 year datasets. Other multi-annual mean plots are formed from all of the complete years of data available for the observational datasets (25 years for ISCCP, 12 years for CERES-EBAF and 5 years for CloudSat/CALIPSO) and 20 year means for the AMIP simulations. We perform a Student's t-test based on inter-annual variability of the data available to determine the 5% significance of model–model and model–observational differences. These have been added to figures in the paper, however in general the inter-annual variability is small compared to the differences discussed.

# 3    Tropical cloud evaluation

Tropics-wide (20$^o$N-20$^o$S) multi-annual average frequency histograms for ISCCP, CALIPSO and CloudSat, together with the outputs from COSP for GA6 and GA7 AMIP experi-

ments are shown in Figure 2a-c. Taking ISCCP first (Figure 2a), retrievals from passive instruments provide a cloud top view. Compared with the newer active instruments, the vertical resolution is poor and there are issues with the height assignment under certain conditions (Mace and Wrenn, 2013). Nevertheless, the optical depth information from ISCCP remains valuable for optical depths greater than approximately 1.0, hence an optical depth frequency profile is also shown. Both GA6 and GA7 tend to simulate too little cloud with intermediate optical thicknesses (1.0-10.0) and slightly too much optically thick cloud. Referring back to the full histograms, this bias appears to be the case for both high and low-top cloud.

Arguably, CALIPSO provides the best global picture of total 2D cloud cover since, unlike the other instruments considered here, it can detect thin sub-visual cirrus. The vertical resolution is good, hence in Figure 2b, as well as providing the full histograms, we collapse along the backscattering ratio axis to provide a vertical profile of cloud frequency. In doing this, for altitudes below 4km we only consider backscattering ratios greater than 5 due to the potential contamination from aerosols in the boundary layer, however above 4km backscattering ratios as low as 3 are included so as to account for very thin cirrus. This choice of the vertical profile of backscattering ratio threshold also gives a profile which most closely matches the CALIPSO cloud detection product used within the RL-GEOPROF dataset (Supplementary material Figure 1). The lidar does become attenuated in the presence of thick ice cloud, and is attenuated quickly in the presence of liquid cloud, hence this profile remains largely a cloud-top view.

Although the CloudSat radar is not sensitive to sub-visual cirrus, it uniquely provides a full 3 dimensional view of the cloud, only becoming attenuated in moderate and heavy rain. Despite the name, it should be noted that CloudSat is sensitive to precipitation as

well as cloud. As for CALIPSO, in Figure 2c we provide a vertical profile of hydrometeor frequency in addition to the full height–radar reflectivity histograms from CloudSat.

Comparing the models with CALIPSO and CloudSat (Figure 2b&c), GA6 clearly has excess amounts of cirrus and this is corrected in GA7. A number of physical improvements included in GA7 have changed the amount of cirrus including the new ice particle size distribution and revised ice optics, however the largest decrease in cirrus has come from the reduction in the rate of cirrus spreading associated with wind shear as the ice falls between successive model levels. This is clear from the orange line on the profile plot of Figure 2b which is a simulation identical to GA6 (the blue line) but with the cirrus spreading reduced to the value used in GA7. The altitude of the cirrus is also too low compared with CALIPSO, but this bias doesn't appear to exist when comparing with CloudSat, which indicates that the issue is associated with very thin cirrus. The CALIPSO histograms indicate that as the cloud thins to the lowest backscattering ratios, the altitude of the cloud should increase, however this does not appear to be the case in GA6. In GA7 the altitude–backscatter ratio relationship is improved such that the highest cloud has the lowest backscattering ratios. This slight increase in the altitude of the cirrus is the result of the revised numerics of the convection scheme. This can be seen from the cyan line in Figure 2b which is a simulation identical to GA6 (the blue line) but with the convection using the 6A scheme (revised numerics). Despite this slight increase in height, the overall altitude of the thin cirrus still remains below that observed by CALIPSO.

The low altitude cirrus bias can be examined in more detail in a case study using a short-range hindcast (Figure 3). In this example (which is typical of other convective cases examined), the A-train overflew a convective system over the South China Sea. The top panels of Figure 3 show the observed and GA6 simulated radar reflectivities. Data

from CALIPSO have been added in locations where the lidar was detecting cloud which was not detected by the radar. It can be seen that the model is able to simulate thin cloud in the upper levels of the convective system right up to the observed altitudes of around 16km. The nominal along-track resolution of the RL-GEOPROF product is 1.7km, so if a threshold of -40dBZ is used for cloud identification and it is regridded onto the model grid, which is 80km near the equator, then an observed cloud fraction over a model grid-box can be estimated. This assumes that the along-track cloud fraction is representative of the 2D grid box. Whilst this is a fair assumption when considering a large number of cases which the A-train will cross at random orientations, there may be an error when considering a single case such as this. The observed and simulated grid-box cloud fraction on the model grid are shown in the lower panels of Figure 3. Large cloud fractions occur up to the top of the convective system in the observations, whereas they reduce quickly above 14km in the model. So it appears that the lack of the highest thin cirrus is primarily because the fractional coverage of grid-boxes is too small in situations where some cloud is present, rather than there being too many completely clear grid boxes at these altitudes. This is likely due to too little condensate being detrained at these altitudes, with what there is being either the result of convection going slightly deeper on occasional timesteps or, more likely, some of the condensate being advected vertically having been detrained below.

Moving down in altitude, Figure 2b suggests the models have too little mid and low top cloud in GA6, whereas Figure 2c may be interpreted as GA6 having considerably too much. However, the excess hydrometeor frequency at lower levels in GA6 is entirely due to excess drizzle in the model rather than cloud. This can be demonstrated by re-running GA6 but not passing the large scale precipitation field to the CloudSat simulator (cyan

line in Figure 2d). In this case the excess hydrometeor fraction is completely removed. Examining these drizzle rates in the model, they are very low (typically $<0.005mm/hr$, not shown), possibly explaining why this model defect had not been spotted before, and again showing the benefit of carrying out evaluation against multiple datasets. This anomalous drizzle is corrected in GA7 to leave the hydrometeor fraction slightly too small at low levels (Figure 2c), which is believed to be mainly due to a lack of heavy convective rain (region of the histogram with radar reflectivities $>0$). The improvement in drizzle in GA7 is entirely due to the warm rain microphysics package, which can be demonstrated if GA6 is run again (all fields passed to the simulator) with just the GA7 change to the warm rain microphysics applied (Figure 2d). Within this package, the change to use the Khairoutdinov and Kogan (2000) scheme reduces auto-conversion rates by a factor of around 100 compared with the scheme in GA6. These rates would be too low without the Boutle *et al* (2014) GCM upscaling, however even after this correction, the auto-conversion rates remain around 10 times smaller than GA6 which accounts for the removal of the spurious drizzle.

Figure 3 shows, under the cirrus shield in GA6, an extensive region of high hydrometeor fraction and reflectivities of the order -10dBZ, between the surface and 7km which is absent in the observed transect. This is consistent with the region of the histogram in Figure 2c where there is spurious large-scale rain. It is likely that large scale cloud is forming in the moist air around the convective system and that it is undergoing auto-conversion, showing up as a strong signal in the CloudSat simulator. In GA7 (not shown) this precipitation signal is removed with just a cloud signal remaining at around -40dBZ. It should be noted that this is in a region largely attenuated for CALIPSO as it is below the cirrus shield and so doesn't contribute to the 'missing' mid-top cloud which is believed to be more

cumulus congestus rather than large-scale.

Tropical low cloud can be more easily assessed if regions are examined in which deep convection is rare/non-existent. Considering a region of the tropical Pacific dominated by trade cumulus and comparing with CALIPSO (Figure 4), GA6 appears to have too little cloud. The forced shallow cumulus scheme improves the amount of shallow cumulus at heights of around 1km, although there looks to be a secondary peak in low cloud around 2km which is absent in both configurations of the model. The region does receive some thin cirrus outflow from nearby deep convective regions, however the amounts are far too large in the model. This indicates that the cirrus lifetime is too great, possibly due to errors in microphysical processes, or macrophysical fields (such as relative humidity being too high). Although improved in GA7 due to the reduced cirrus spreading rate, the excess cirrus in this region remains. The observed cloud fraction is sensitive to the effective resolution used by the retrieval algorithm. GOCCP uses the instrument nominal resolution, whereas RL-GEOPROF uses a minimum averaging length of 5km to improve the signal-to-noise ratio. Chepfer *et al* (2013) show that the averaging effect is sensitive to the length averaged and is higher for low-level, small-scale broken cloud. For high clouds, the differences between GOCCP and the CALIPSO cloud retrieval used by RL-GEOPROF are dominated by the SR detection threshold. The height-dependent SR detection threshold used in this study increases the sensitivity to high clouds (supplementary Figure 1). For cirrus clouds in the regions shown in Figure 4, the bias in cloud fraction introduced by lack of averaging in GOCCP is smaller than 0.05 (Figure 10 in Chepfer *et al*, 2013), supporting the interpretation that the cirrus amounts simulated by the models are excessive in this region.

Over the past couple of decades, a key focus of model development in the UM in

relation to clouds has been on improving the simulation of subtropical stratocumulus due to its importance in determining the global cloud feedback under climate change (e.g. Bony and Dufresne, 2005). Many models have too little cloud in this region, with what there is being too bright (Nam *et al*, 2012). A number of improvements in previous configurations have resulted in the cloud amounts being in very good agreement with CALIPSO (Figure 4), although the low cloud amounts are reduced slightly in GA7 as a result of the change in the aerosol scheme to GLOMAP-mode. Compared with ISCCP, GA7 has considerably too little moderately reflective cloud in this region, but slightly too much optically thick cloud indicating that what cloud there is remains too reflective. Consistent with this, comparison against a number of observational datasets indicates that the cloud effective radius simulated by the model is too low in many regions, including subtropical stratocumulus (not shown), and is indicative of the aerosol cloud indirect effect being too strong (Walters *et al*, 2017).

Compositing cloud data by large scale variables is a useful way of summarising the tropical cloud structures across different meteorological situations. The most common are to composite against $500hPa$ vertical velocity (Bony *et al*, 2004) and a measure of lower tropospheric stability. A number of measures for the latter have been proposed (e.g. Klein and Hartmann, 1993; Williams *et al*, 2006; Wood and Bretherton, 2006), however here we simply use the spacial variation in sea surface temperature (SST) (e.g. Williams *et al*, 2003). We composite the observed and modelled CALIPSO cloud profile by daily $500hPa$ vertical velocity ($\omega_{500}$) and SST (Figure 5). The desirable increase in altitude of the cirrus, discussed above for the tropics as a whole, can be seen in all the large scale vertical velocity regimes. The reduced cirrus amount in Figure 2b is also reflected in Figure 5, with the largest reduction in regions of strongest ascent. However there now

appears to be too little cirrus in weakly ascending regimes in GA7. This separation by regime therefore gives useful insights on where there might be compensating errors in the tropical mean picture provided by Figure 2.

The SST composites appear to better separate the stratocumulus regions at the coldest end as these bins clearly show higher fractions of boundary layer cloud. There is slightly too little low cloud in a number of the SST and $\omega_{500}$ composite bins, whilst there looks to be too much stratocumulus in the coldest SST bin. However in general, low-top cloud amounts appear to be reasonably well simulated.

# 4  Cloud evaluation in the mid-latitude storm tracks

The weather over the mid-latitude oceans is characterised by the passage of synoptic systems. Since the cloud structures change on a daily basis, compositing of climatological data is essential. Here we follow Govekar *et al* (2011) to analyse RL-GEOPROF cloud data around a composite cyclone, using the cyclone compositing technique of Field and Wood (2007). Cyclone centres are identified from daily ERA-I PMSL (pressure at mean sea level) data over the northern hemisphere oceans ($35^oN$-$70^o$) and the RL-GEOPROF data extracted for a $30^o$ latitude by $60^o$ longitude box centred on the cyclone. All the cyclones from 5 years worth of daily December-January-February (DJF) data are then averaged to form a composite cyclone. In order to visualise the composite, Figure 6 shows several sections through the 3 dimensional composite. The top panels are horizontal sections in the boundary layer (1.7km) and upper troposphere (6km) with the mean PMSL contoured. The positions of frontal features will vary with time and between systems, and the size of cyclones varies which also smooths the composite, but on average it would be expected that fronts would occupy the south-east quadrant with a cloud head wrapping around the

north of the cyclone (Field and Wood, 2007). This can be seen as higher cloud fractions in these locations in the section at 6km, whilst the boundary layer hydrometeor fraction appears more symmetrical around the cyclone with a maximum near the centre. The lower panels on Figure 6 are vertical sections across the composite to the south and to the east of the centre, with the contours indicating the average vertical velocity from ERA-I (dashed indicates ascent). The east-west cross section at $4^o$ south of the centre has large-scale descent in the cold air on the left of the plot with cloud largely confined to the boundary layer. Moving to the east, there is a change to large scale ascent and higher cloud fractions throughout the troposphere as we cross the composite warm conveyor belt. The north-south section shows similar strong ascent and high cloud fractions in the cloud head just to the north of the surface cyclone centre, but also an indication of a secondary maximum at the southern end ($-5^o$, 2km to $-12^o$, 6km) where the section will sometimes pass through a trailing cold front.

The same compositing methodology can be applied to the model with a simulated RL-GEOPROF product from the CloudSat and CALIPSO simulators. The difference between the modelled and observed composite cyclones can be calculated (Figure 7). Both model configurations have excess hydrometeor frequency in the boundary layer around the cyclone. This is slightly improved in GA7 with the largest bias confined to the western periphery of the cyclone. GA6 also has considerably too much cirrus on the rearward side of the frontal regions. The excess cirrus is completely removed in GA7 through the reduced cirrus spreading rate such that cloud amount biases in the free troposphere around the GA7 composite cyclone are very small.

A case study again provides a useful illustration of the excess cirrus in GA6 (Figure 8). In this example the A-train passed over a mature depression in a very similar section to

the lower-right panel of the cyclone composite in Figure 6. Given this is a forecast with a greater than 1 day lead time, the simulated positions of the frontal features are very good. The main bias is the width of the cloud associated with the warm conveyor belt being too large, especially visible for the trailing cold front at around $44^oN$. Examining the cloud fraction on the model grid, there are instances on the edges of the fronts where the observations suggest clear sky but the model simulates partially cloud grid-boxes. In contrast, within the cloud head around $60^oN$ there is an indication that the model too readily breaks up the cloud when the grid box should be completely covered. This tendency for the model to too often simulate partially cloudy grid-boxes rather than 0% or 100% is consistent with previous experience with the UM (e.g. Mittermaier, 2012) and may relate to a critical relatively humidity still be used to initially form/decay cloud when the grid box is 0%/100% cloud covered respectively.

The same cyclone compositing methodology has been carried out over the northern hemisphere oceans for June-July-August (JJA) and for the summer and winter seasons in the southern hemisphere ($40^oS$–$70^oS$). We have also composited anticyclones using the same cyclone settings as Field and Wood (2007), but testing for $d^2p/dx^2 + d^2p/dy^2 < 0$ in order to identify a local maximum in surface pressure rather than a local minimum. All the plots are available in the Supplementary Material and show a broadly similar picture of excess cloud in the free troposphere and boundary layer in GA6, the former being essentially fixed and the latter improved in GA7. The GA6 cirrus biases in anticyclones are smaller than cyclones, but the boundary layer errors are more similar. The cyclone composite for the Southern Hemisphere summer now suggests slightly too little mid-level (2-5km) cloud on the cold air side (poleward and westward side) of the cyclone in GA7 (Supplementary Material Figure 2). This may be associated with a lack of congestus cloud

here which is a long-standing problem, but was being masked in GA6 through the excess cirrus throughout the free troposphere. Govekar *et al* (2011) provided an evaluation of cyclone composite cloud amounts over the Southern Ocean in an earlier configuration of the UM (Australian Community Climate and Earth System Simulator, ACCESS1.3). They concluded that while the cloud simulation was in reasonable agreement with observations, the large scale vertical velocity was poor and they cautioned that there may be a compensating error in the cloud simulation. In both GA6 and GA7, the vertical velocities in the cyclone composites compare well with ERA-I (e.g. Figure 7), hence this issue is no longer of concern.

Despite the cloud amount composites showing cloud fraction errors of less than 0.15 (and often less than 0.05) in GA7, composites of the top of atmosphere (TOA) radiation biases reveal some issues (Figure 9). The OLR is slightly too low across the cyclone composites which is believed to generally reflect a slight tropospheric cold bias in the model. However, the main issue is in the reflected shortwave (RSW). Unsurprisingly, this error is larger in the summer season in each hemisphere when the insolation is greatest. The northern hemisphere has excess RSW across the cyclone composite, and particularly in regions of the composite with more cloud. In contrast the southern hemisphere has a large deficit of RSW on the cold air side of the cyclone, a common bias in climate models (Bodas-Salcedo *et al*, 2014). The northern hemisphere being too reflective can also be seen in the anticyclone composites (Supplementary material Figure 4), but the southern hemisphere error seems mainly confined to the cyclone composite.

Figure 10 shows composite cyclone in-cloud albedo biases against ISCCP. In contrast to the RSW, the in-cloud albedo does not depend on the insolation and so a cloud microphysical error affecting the albedo which is present throughout the year will appear

the same in the DJF and JJA plots. However, these albedo biases have a structure which is consistent with the radiation errors e.g. the fact that the negative RSW bias on the poleward side of the southern hemisphere cyclone is larger in DJF than JJA is partly due to there being a larger albedo error in the austral summer rather than just the insolation being higher. In the northern hemisphere, the DJF in-cloud albedo has the largest positive bias in the south-west quadrant of the composite cyclone, which is where there is the largest positive bias in RSW; whereas in JJA, the in-cloud albedo bias is more in the central and south-east side, again consistent with the RSW error. Unlike the in-cloud albedo errors, the cloud amount errors in Figure 7 and the Supplementary Material appear not well correlated spatially with the RSW errors around the composite cyclone. We therefore suggest that microphysical processes are primarily responsible for the SW errors through incorrect cloud albedos. This is a good example of the value of the compositing technique for understanding the likely cause of radiation errors. Although the subject of ongoing research, we believe that the negative in-cloud albedo bias on the cold-air side of the southern hemisphere cyclone is due to a lack of super-cooled liquid water (Bodas-Salcedo *et al*, 2016), whereas the northern hemisphere bias is thought to be associated with issues around the simulation of aerosols and their interaction with the clouds, particularly the strong cloud–aerosol interaction noted earlier.

# 5 Cloud evaluation over mid-latitude land

Much of the northern hemisphere mid-latitudes are land covered and here we composite the RL-GEOPROF hydrometeor fraction and CALIPSO cloud fraction, along with their simulated equivalents, by $\omega_{500}$. We illustrate the results for DJF (Figure 11), although JJA is qualitatively similar. The excess cirrus issue in GA6 can again be seen and this

is removed in GA7. For some of the regimes, it looks as though there may be now too little cirrus in GA7, although these are the relatively less populated regimes of strongest ascent and strongest subsidence.

There appears to be a significant excess of hydrometeor fraction in both model configurations at around 1km, however the CALIPSO profiles suggest the cloud fractions at this level are generally correct. This exemplifies the utility of using multiple observation types and indicates that the excess hydrometeor in the RL-GEOPROF comparison is either low cloud in situations where there is thick high cloud above, and/or excess precipitation. Although a detailed investigation is yet to be carried out, it is suspected that both may be contributing. Case study analysis in the vicinity of the UK in February 2015 has identified a few occasions with spurious drizzle/light rain falling from stratocumulus (not shown). Unlike the warm drizzle cases in the tropics which were improved by changes to the auto-conversion scheme in GA7, these mid-latitude winter cases have frozen cloud tops. It is possible that the microphysical errors leading to excess drizzle in frozen stratocumulus seen in the case study are a general issue contributing to the bias in Figure 11. However, low cloud is frequently simulated by the model over land areas in the winter and given that a cirrus shield is present on many occasions, it is quite possible that excess low cloud is also being simulated but shielded from the CALIPSO simulator.

The active satellite instruments provide an invaluable global picture of the three dimensional cloud structure through most of the troposphere, however the radar can be contaminated with ground clutter in the lowest few hundred metres, and the lidar will frequently be attenuated before detecting the lowest cloud layers. Accurate predictions of cloud near the surface are of the highest importance for a number of users of the model, especially aviation. Here we use SYNOP data which, whilst having a reasonable global

coverage over land, are likely to be the most reliable observation type available for this lowest layer. They avoid the ground clutter issues of remote sensing from space and an upward pointing ceilometer or human observer looking from the ground is likely to achieve higher accuracy for low cloud bases as they avoid the problem of attenuation from cloud above. By looking at the lowest 1km, many of the issues associated with the SYNOP data (combining human and automated data and differing observational errors associated with each) will be minimised (Mittermaier, 2012). In order to confine the analysis to cloud with bases below 1km, we use the cloud base height observation and look at frequency of occurrence of cloud bases below 1km. The cloud base height is defined as the height of cloud with coverage of 3 oktas or more, hence instances of small cloud coverage are excluded from this analysis. As a consequence, significant model biases in this diagnostic can appear if the observed cloud amount is typically just over 3 oktas and the model cloud fraction is just under (or vice-versa). This appears to be an issue for the UM in parts of the tropics where too little shallow cumulus is simulated and typically the model has cloud fractions of <3 oktas (i.e. grid box fraction of <0.375) whereas fractions over this threshold are often observed and hence a cloud base height assigned. More generally the diagnostic is reflecting errors in the frequency of occurrence of low-base cloud. Based on comparison with the active instruments at higher altitudes, we suspect that biases are more often reflecting errors in the frequency of occurrence of low cloud rather than errors in the cloud base height on any one occasion.

Figure 12 shows the day 1 bias in the frequency of occurrence of cloud base height for one year of data since GA6 became operational. Note that here the term 'bias' uses the definition of the the international Joint Working Group for Forecast Verification Research as being (hits + false alarms)/(hits + misses) (http://www.cawcr.gov.au/projects/verification/),

so a value of 1.0 would indicate no model bias. In order to visualise the station density more clearly, we show a section over Europe which illustrates the key points of the mid-latitude land regions in general. Over most of the area the model performs well and is essentially unbiased. Its performance over the UK is comparable to a 1.5km convective permitting configuration of the UM which is run operationally over the region (not shown). However over areas of notable orography, such as the Alps, there appears to be excess low cloud in the model. In contrast, around some of the coasts (especially France and Italy) there is too little low cloud. Further work is required to identify the cause of these errors.

# 6 Global cloud radiative effects

Traditionally the primary evaluation of clouds in climate models was through an assessment of their impact on the TOA radiation budget. However, as discussed in the introduction, this could hide compensating errors which might result in an incorrect cloud radiative response to climate change. We suggest instead that this assessment should be towards the end of a wider cloud evaluation, such as that presented above, feeding into the model development process.

The GA6 and GA7 bias in TOA RSW and OLR is shown in Figure 13. Generally the biases are reasonably similar with some local improvements (e.g. in RSW over India and the equatorial Indian Ocean) and local detriments (e.g. in OLR over the Maritime Continent). A widespread bias for the free troposphere to be too cold in GA6 has been slightly improved in GA7 (mainly due to the introduction of the 6A convection scheme (Walters *et al*, 2017)) which largely accounts for the general increase in OLR in the newer model. Given that GA7 will be the physical model underpinning the UK submission to CMIP6, it

is useful to compare back to HadGEM2-A (Hadley Centre Global Environmental Model 2 - Atmosphere; Martin *et al*, 2011) which was the CMIP5 submission. It should be noted that HadGEM2-A is a comparatively old model with some 7 years of continuous model development having occurred between this and GA6, hence the differences in the radiation budget are much larger. It can be seen that GA7 is a considerable improvement on HadGEM2, especially for the RSW. The error in the sub-tropical cumulus transition regions of excess RSW has been removed and there is now a smaller negative bias in GA7. The lack of RSW over the Southern Ocean has been reduced by a third and RSW & OLR biases over the Maritime Continent have been significantly improved.

Metrics are often used to summarise the overall performance of the model. There are few such metrics in the literature for NWP–seasonal cloud prediction applications, however a number have been proposed for aspects of the cloud simulation which are likely to be important for the radiative response of cloud to climate change (e.g. Pincus *et al*, 2008; Klein *et al*, 2013; Myers and Norris, 2015). Here we illustrate the calculation of metrics as the final step in the evaluation process by presenting the present day Cloud Regime Error Metric (CREMpd) of Williams and Webb (2009). This metric assesses the ability of the models to simulate primary cloud regimes (as determined by the daily mean cloud cover, optical depth and cloud top height) with the correct frequency of occurrence and radiative properties. Here we modify one aspect of the Williams and Webb (2009) approach by using the newer global regimes proposed by Tselioudis *et al* (2013) instead of calculating the tropics, extra-tropics and snow/ice covered regions separately. Figure 14 shows the CREMpd for GA6, GA7 and all the CMIP5 models for which the required data are available, with zero being a perfect score compared with the observations. GA6 is comparable with the previous HadGEM2-A model as being among the better performing

models on this metric, with GA7 performing slightly worse but still competitive with other CMIP5 models. Having a climate change application focus, CREMpd is very sensitive to the accuracy of the simulation of clouds with the strongest net radiative effect, namely stratocumulus. Consequently GA7 is penalised compared with GA6 for the overall reduction in the albedo of sub-tropical stratocumulus (Figure 4). In contrast, the metric has limited acknowledgment of the large improvements in the amount of cirrus in GA7 since the radiative effect of this, largely sub-visual cloud, is small.

# 7 Summary and discussion

In this study we have attempted to convey a more thorough evaluation of cloud than has traditionally been undertaken as part of a model development process. Our experience has been that using a limited set of diagnostics and/or observational datasets can result in compensating errors. An example is the rate of cirrus spreading which was part of a change introduced in GA4 (Walters *et al*, 2014), but at the time we were not routinely evaluating against CALIPSO. We have now discovered that this was producing excessive amounts of sub-visual cirrus and this has been corrected in GA7. The ability to compare the models with multiple satellite datasets using COSP, combined with a variety of compositing techniques has permitted a detailed, process-orientated evaluation to be undertaken. We find that the use of multiple datasets and diagnostic techniques to draw a consistent picture of model errors is likely to reduce the risk of drawing the wrong conclusions and more accurately focus future model development. Examples include the comparisons between CloudSat and CALIPSO that demonstrate errors due specifically to thin cirrus, or to excess precipitation as opposed to cloud error; the use of cyclone composites of cloud amount, in-cloud albedo and radiative fluxes to show, through similar spatial patterns,

that the error in the RSW is likely due to errors in the in-cloud albedo rather than cloud amount; the use of surface-based observations for the lowest atmospheric layers where remote sensing from space becomes problematic; etc.

The combination of CloudSat and CALIPSO provides a unique three dimensional observational dataset of hydrometeor frequency through much of the atmosphere. We find that some care is required in its use for model evaluation in terms of separating cloud and precipitation, and the ability to perform multiple simulations passing different fields to the simulator can be valuable. Despite being an older satellite dataset, the optical depth information from ISCCP remains extremely valuable for model evaluation purposes. Evaluation of very low cloud (<1km) remains a challenge, especially when thicker cloud exists above. We have made use of the SYNOP data which have reasonable coverage over land and, for cloud at these altitudes, may be regarded as fairly reliable. The thresholds and variables available in the SYNOP data do limit the evaluation though.

A key part of our evaluation process is the cross-timescale assessment which enables the statistical robustness of the climate simulations to be combined with more detailed analysis of case studies in NWP hindcasts to understand the model errors at the process level. Although many centres don't routinely run simulations across these timescales, the AMIP and Transpose-AMIP experiments proposed by the Working Group Numerical Experimentation (WGNE) provide a relatively simple methodology enabling all centres to benefit from this approach.

GA6 generally performs well given the critical examination presented here. The main errors are:

1. A considerable excess of thin, often sub-visual, cirrus erroneously extending from thicker cirrus clouds which ought to be present. This has been essentially fixed in

GA7.

2. In-cloud albedo is too high in tropical and extra-tropical stratocumulus, except on the cold air side of cyclones in the Southern hemisphere where they are too low.

3. A slight excess of boundary layer hydrometeor fraction over the mid-latitudes which is suspected to be a combination of excess cloud and drizzle.

Apart from errors in external driving factors such as the location and timing of convection and synoptic systems, item 2 in the list above is the main cloud error affecting the mean radiation bias.

Although we have attempted the most comprehensive assessment possible in the time available, the task is inevitably open ended. The main omissions which we would have liked to address are an evaluation of the diurnal cycle of clouds globally and cloud over high latitude regions. Sea ice and snow cover are likely to be quite sensitive to cloud and this is a region which has generally received little detailed systematic cloud evaluation. Use of data from additional instruments such as ground-based cloud radar and lidar, and from the Multi-angle Imaging Spectro-Radiometer (MISR) satellite instrument would also be valuable additions in future studies.

# Code availability

The UM is available for use under licence. A number of research organisations and national meteorological services use the UM in collaboration with the Met Office to undertake basic atmospheric process research, produce forecasts, develop the UM code and build and evaluate Earth system models. For further information on how to apply for a licence see http://www.metoffice.gov.uk/research/collaboration/um-collaboration. Versions 8.6

684  (for GA6) and 10.3 (for GA7) of the source code are used in this paper.

# Acknowledgments

686  This work was supported by the Joint DECC/Defra Met Office Hadley Centre Climate
687  Programme (GA01101). We thank Cyril Morcrette, Paul Field and David Walters for
688  useful discussions throughout the study.

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

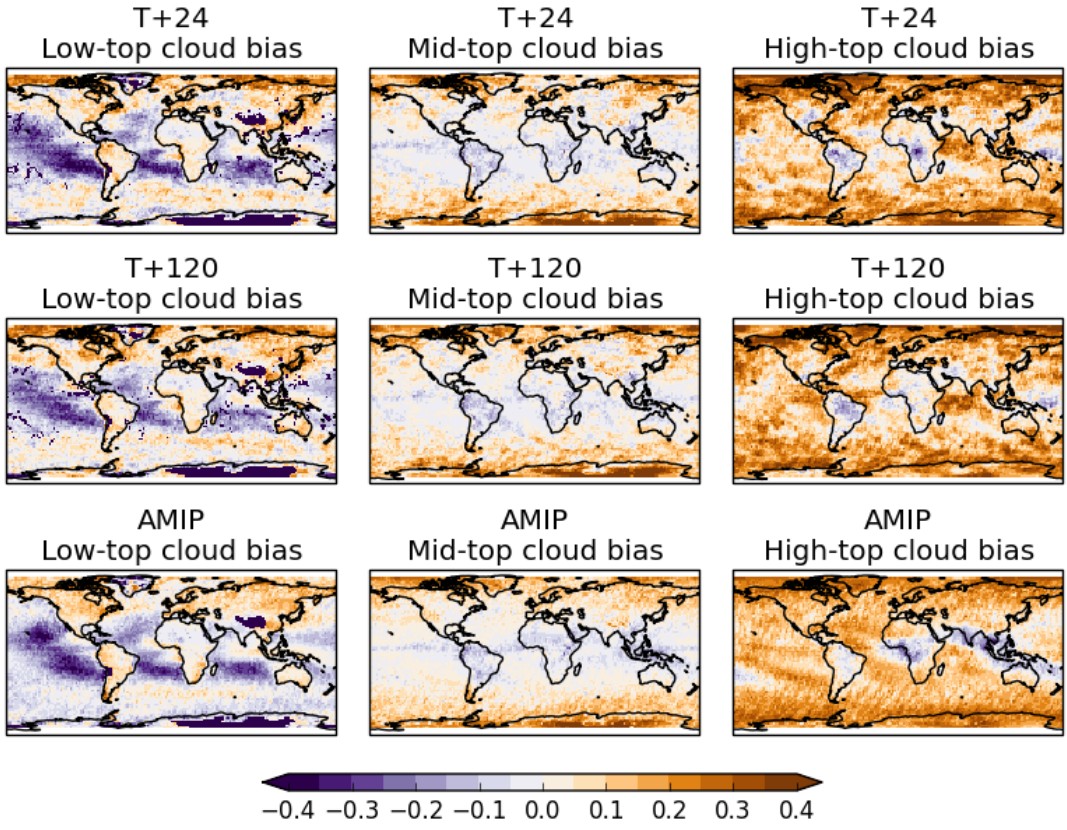

Figure 1: Absolute bias (model field minus observed field) in GA6 configuration of the UM for low (left) mid (centre) and high (right) fractional cloud cover against the GCM Orientated CALIPSO Cloud Product (GOCCP), using the CALIPSO simulator in COSP (see Section b. Top and middle rows are mean biases at day 1 and day 5 averaged across all the NWP hindcasts at N320 (40km in mid-latitudes) resolution. The bottom row is the bias in the AMIP climatology at N96 (135km in mid-latitudes) resolution.

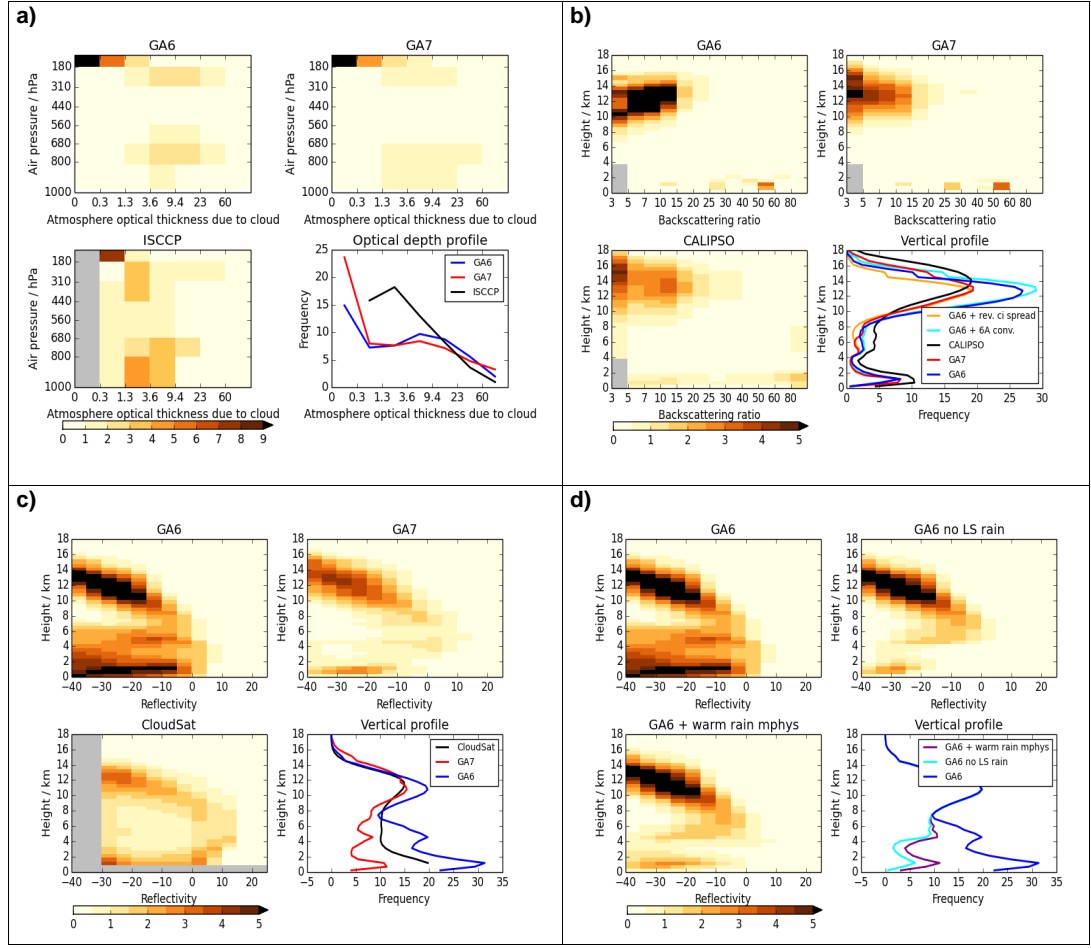

Figure 2: Tropical multi-annual mean observed and GA6 & GA7 simulated satellite data summaries. a) ISCCP cloud-top pressure–optical depth joint frequency histograms. Lower right panel is a single optical depth frequency histogram (i.e. the joint histograms have been summed across cloud top pressure bins). The threshold optical depth for detection by ISCCP is believed to be approximately 0.3, hence the masking of the lowest bin in the observed histogram. b) CALIPSO backscattering ratio frequency histograms by height. Lower right panel is a single height frequency histogram (i.e. the backscattering ratio histograms have been summed across backscattering ratio bins in each height bin). Within the boundary layer, backscattering ratios <5 are likely to be due to aerosols (see Supplementary Material Figure 1) and hence are masked. The lower right panel also shows frequency profiles for GA6 with the cirrus spreading reduced to GA7 values, and GA6 but with the 6A convection scheme used. c) CloudSat radar reflectivity (dBZ) frequency histograms by height. Lower right panel is a single height frequency histogram (i.e. the reflectivity histograms have been summed across reflectivity bins in each height bin). d) As c) but showing GA6, GA6 without large-scale rain being passed to the simulator, and showing GA6 plus the warm rain microphysics package which is included in GA7. Colour scale for the histograms show frequency of occurrence of cloud/hydrometeor in the bin (%). Shading around the line plots has been added to reflect significance bounds, however this is often less than the thickness of the plotted lines.

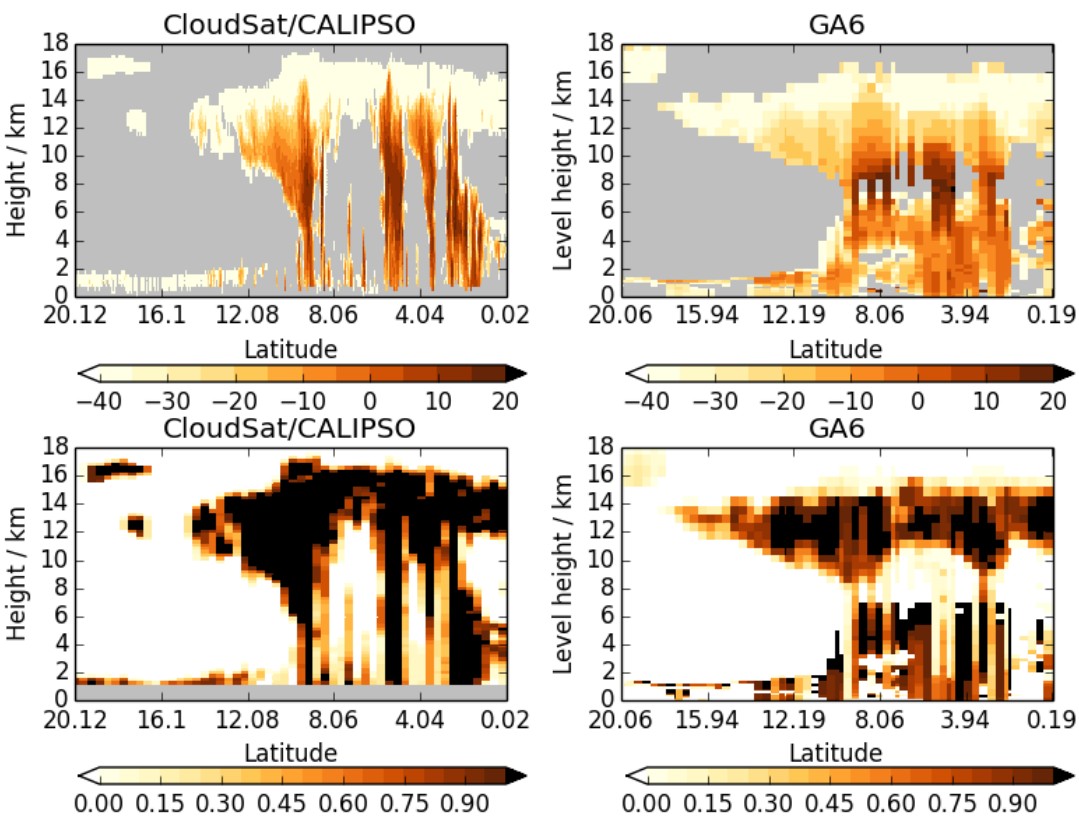

Figure 3: Case study of a GA6 6 hour forecast verifying at 18:00UTC on 17th December 2010 for an A-train pass over the South China Sea. Top: the observed and simulated radar reflectivities (dBZ) with situations in which the lidar detected cloud but the radar did not being included with a nominal value of -40dBZ (e.g. Mace and Wrenn, 2013). Bottom: observed and simulated cloud fraction on the model grid.

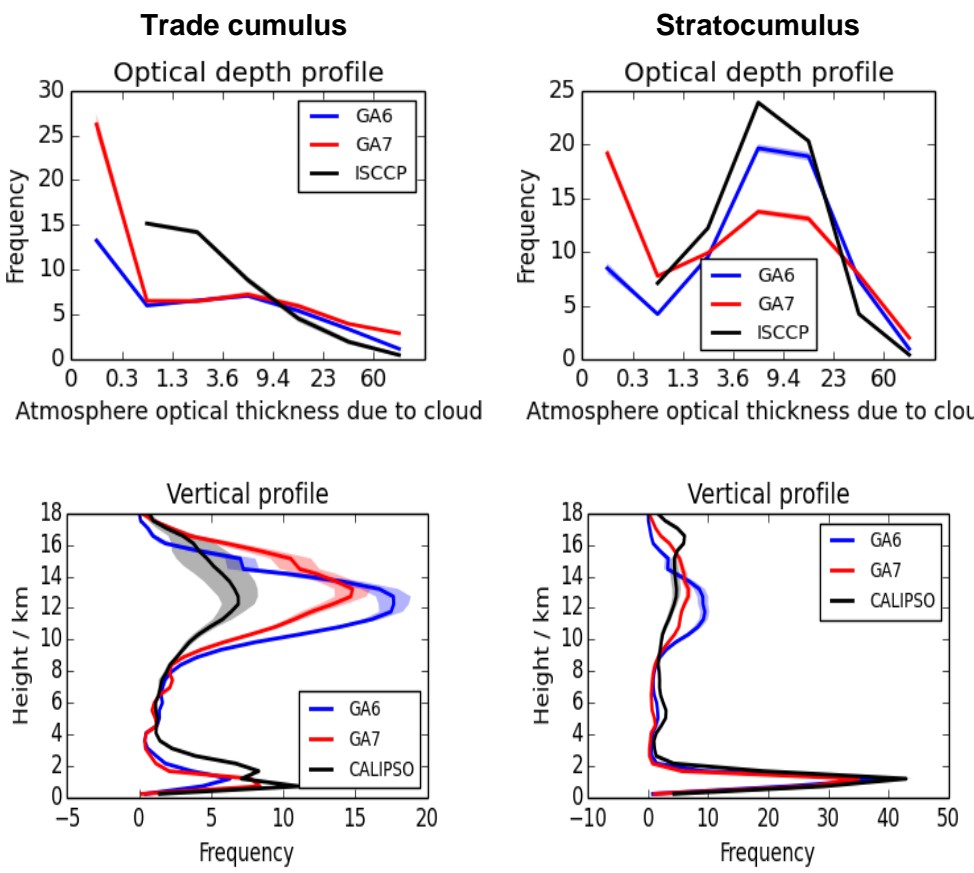

Figure 4: Observed and simulated multi-annual mean ISCCP optical depth frequency histograms (top) and CALIPSO height frequency histograms (bottom) for a trade cumulus region (130-160ºW, 0-20ºS, left) and stratocumulus region (80-90ºW, 0-20ºS, right). Shading around the line plots has been added to reflect significance bounds, however this is sometimes less than the thickness of the plotted lines.

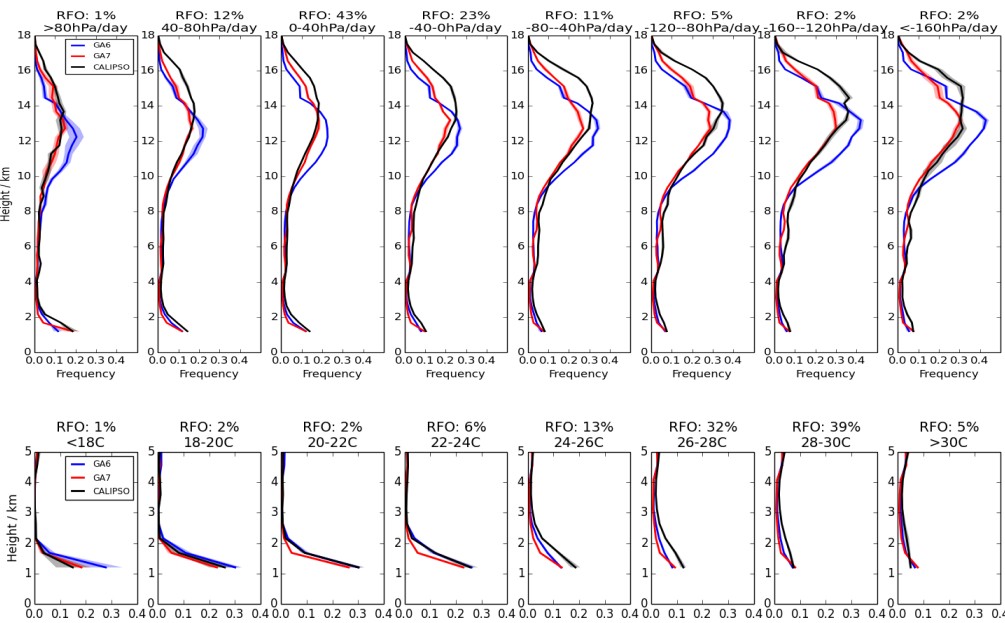

Figure 5: Observed and simulated CALIPSO height frequency histograms composited by daily $\omega_{500}$ (top) and SST (bottom) over the tropics (20ºN–20ºS). Only the region below 5km is shown in the lower plot to focus on low cloud. The range and relative frequency of occurrence (RFO) are shown at the top of each bin. Negative $\omega_{500}$ indicates ascent. Shading around the line plots has been added to reflect significance bounds, however this is sometimes less than the thickness of the plotted lines.

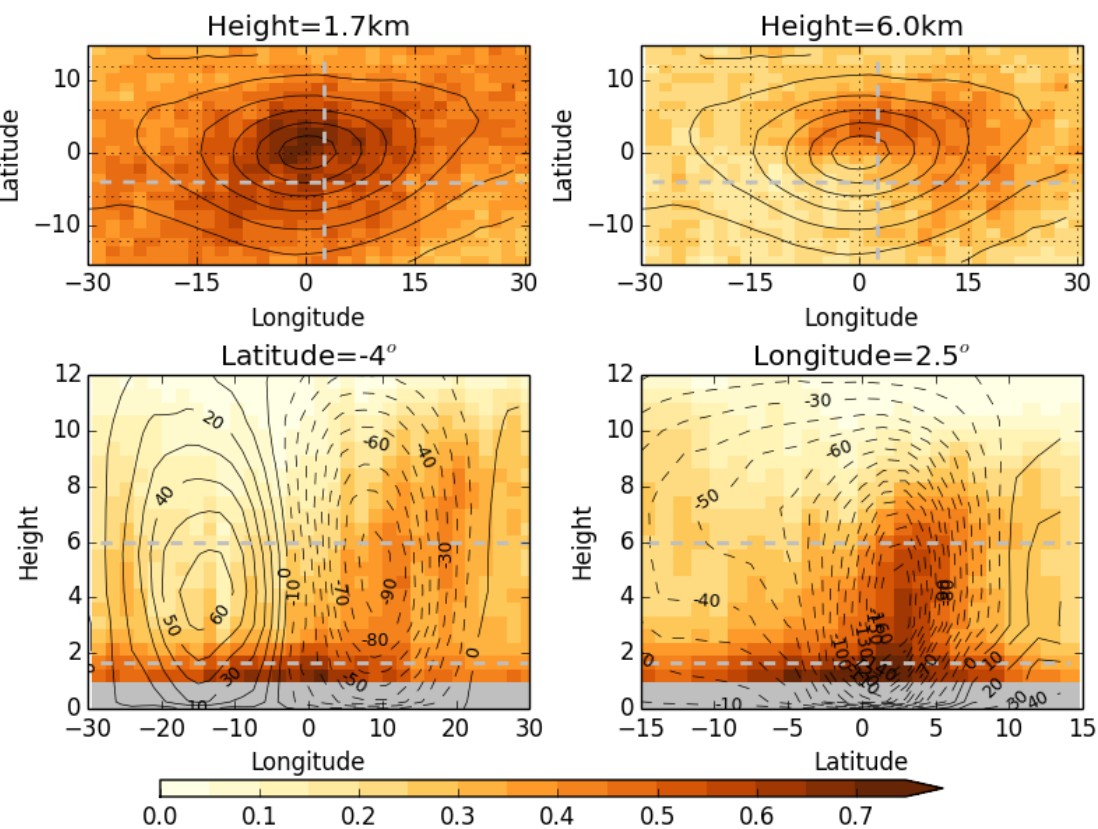

Figure 6: Distribution of average observed hydrometeor (cloud plus precipitation) fraction (colours) around a composite of ERA-I cyclones over northern hemisphere oceans for 5 years of DJF daily data. Top row shows horizontal sections through the composite cyclone at 1.7 & 6km with the mean PMSL contoured at 4hPa intervals. Bottom row shows vertical sections along the grey dashed lines shown in the top plots. Contours on the lower plots are mean vertical velocity from ERA-I (hPa/day; negative values indicate ascent and these contours are dashed).

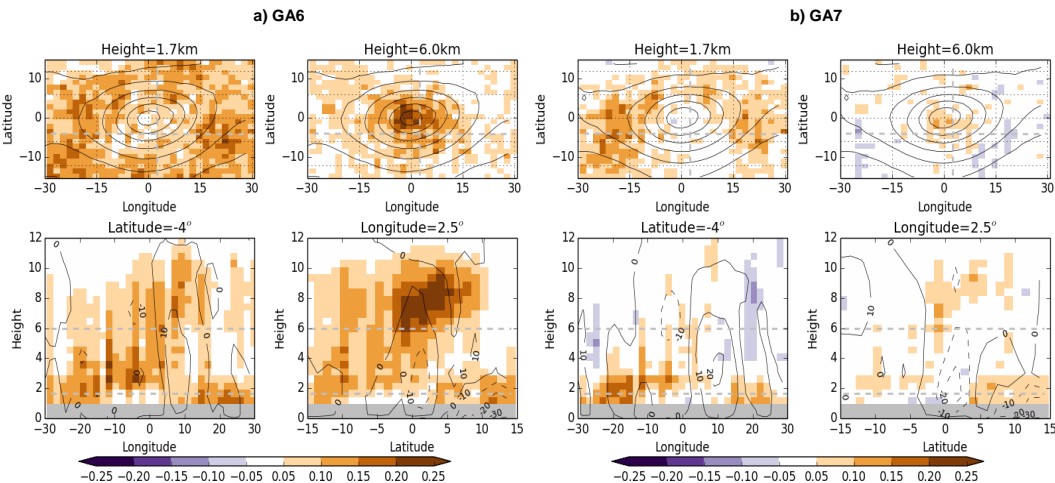

Figure 7: Cloud fraction absolute bias (model field minus observed field) (colours) for composite cyclones. Produced as per Figure 6 for a) GA6 b) GA7 and the observed composite then subtracted. Black contours in top plots are the model mean PMSL and in the lower plots are the bias in vertical velocity. Student's t-test based on internnual variability show that errors greater than 0.05 are statistically significant at the 0.05 level.

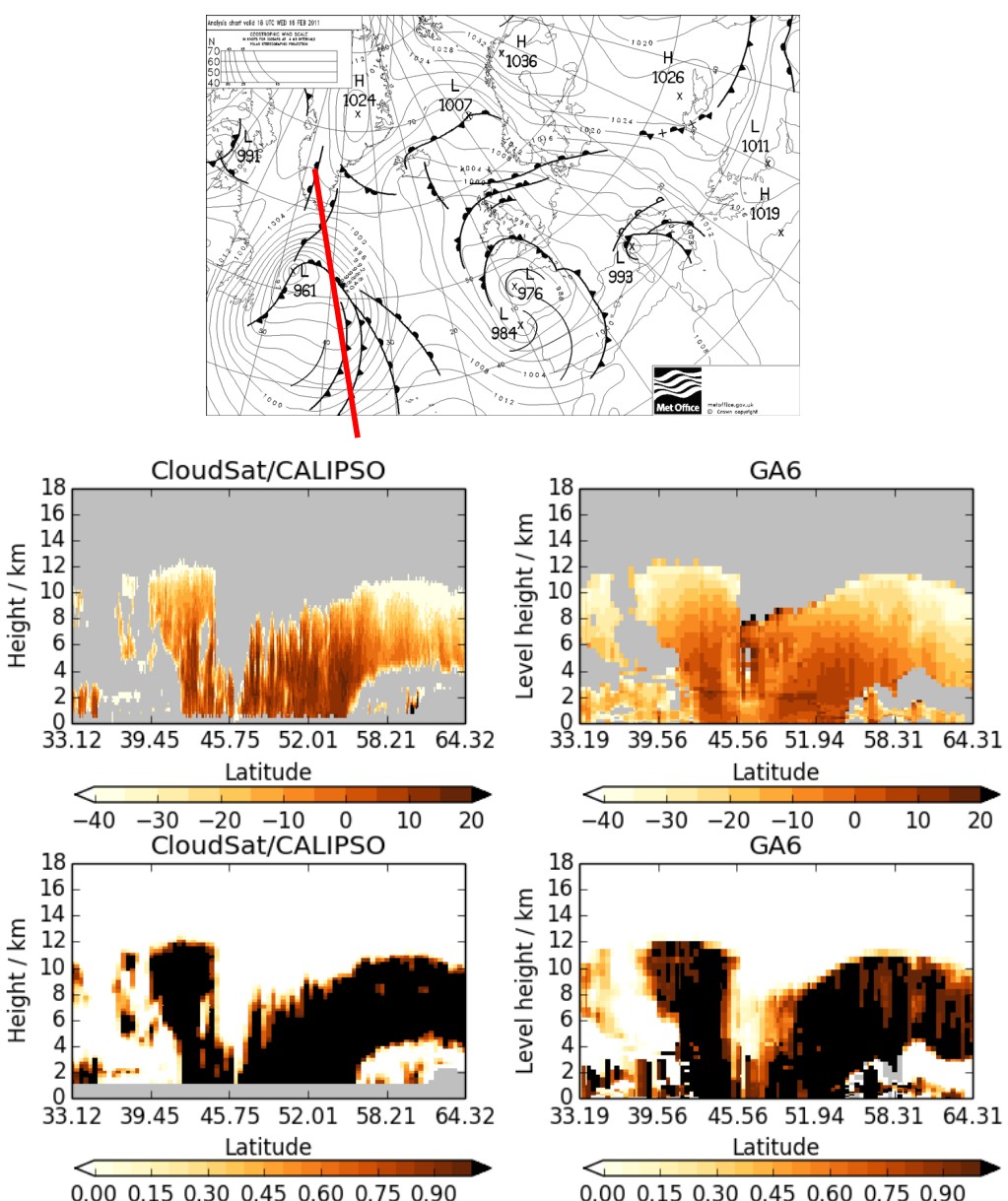

Figure 8: Case study of a GA6 27 hour forecast verifying at 15:00UTC on 16th February 2011 for an A-train pass over the North Atlantic as shown by the red line on the synoptic analysis. Top: the observed and simulated radar reflectivities (dBZ) with situations in which the lidar detected cloud but the radar did not being included with a nominal value of -40dBZ. Bottom: observed and simulated cloud fraction on the model horizontal grid.

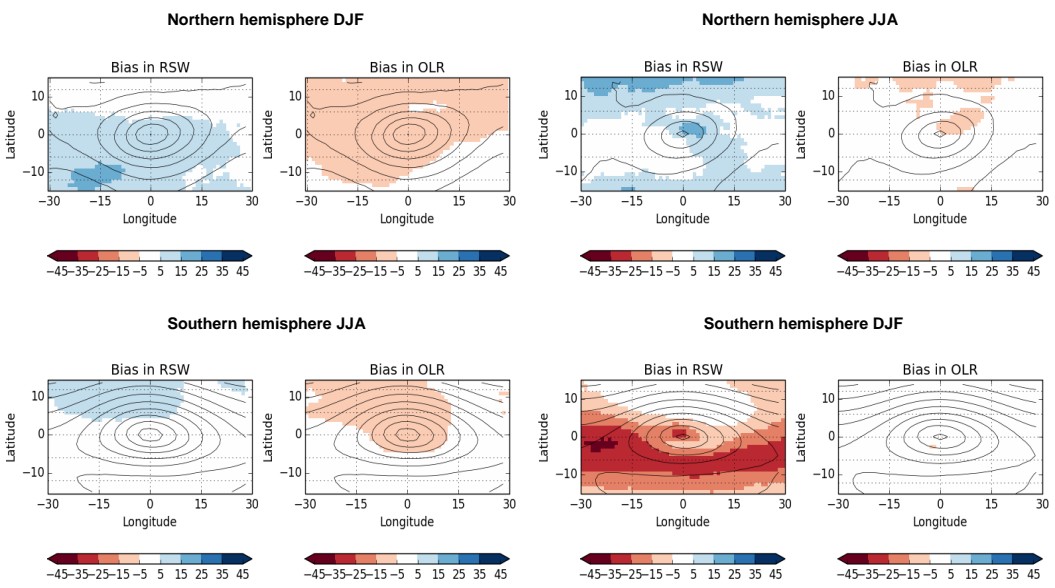

Figure 9: Cyclone composite GA7 mean bias in RSW and OLR (Wm$^{-2}$) against CERES-EBAF (colours). Black contours are GA7 PMSL. Northern and Southern hemisphere composites are shown for the respective winter (left) and summer (right) seasons. Student's t-test based on internnual variability show that errors greater than $5Wm^{-2}$ are statistically significant at the 0.05 level.

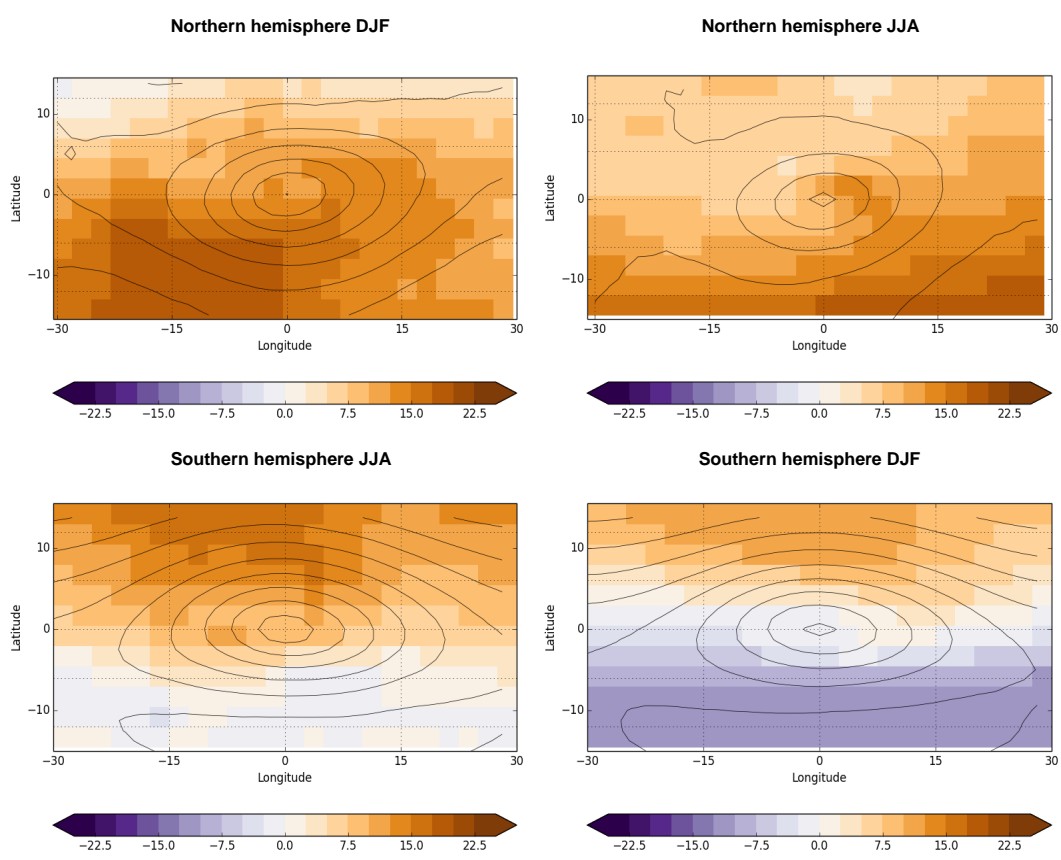

Figure 10: Cyclone composite GA7 mean bias in in-cloud albedo (%) against ISCCP (colours). Black contours are GA7 PMSL.

Figure 11: Observed and simulated RL-GEOPROF and CALIPSO height frequency histograms composited by daily $\omega_{500}$ over northern hemisphere land (polewards of $20^oN$) during DJF. The range and relative frequency of occurrence (RFO) are shown at the top of each bin. Shading around the line plots has been added to reflect significance bounds, however this is sometimes less than the thickness of the plotted lines.

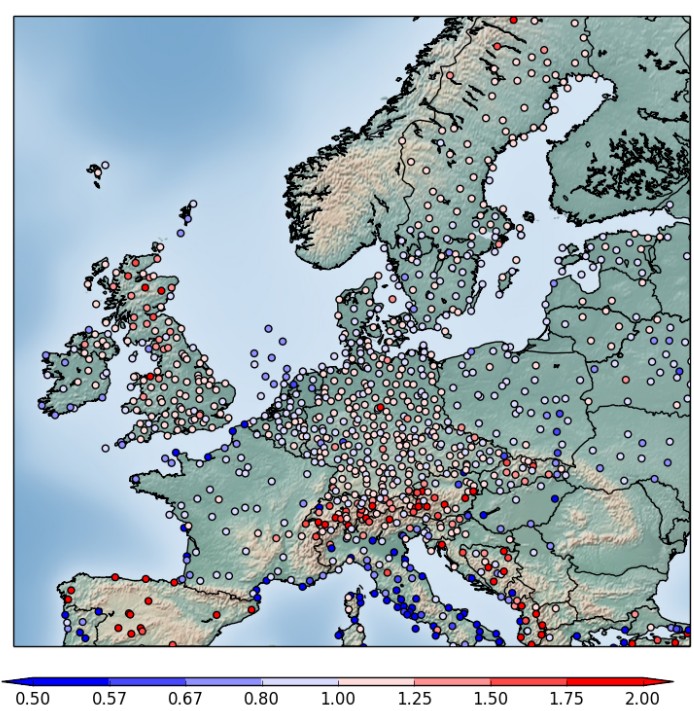

Figure 12: Frequency bias ((hits + false alarms)/(hits + misses)) of cloud base height <1km for cloud fraction ≥ 3 oktas in GA6 against surface station data. The mean bias of 6-hourly forecasts between 16th July 2014 and 15th July 2015 at a 24 hour forecast lead time are shown.

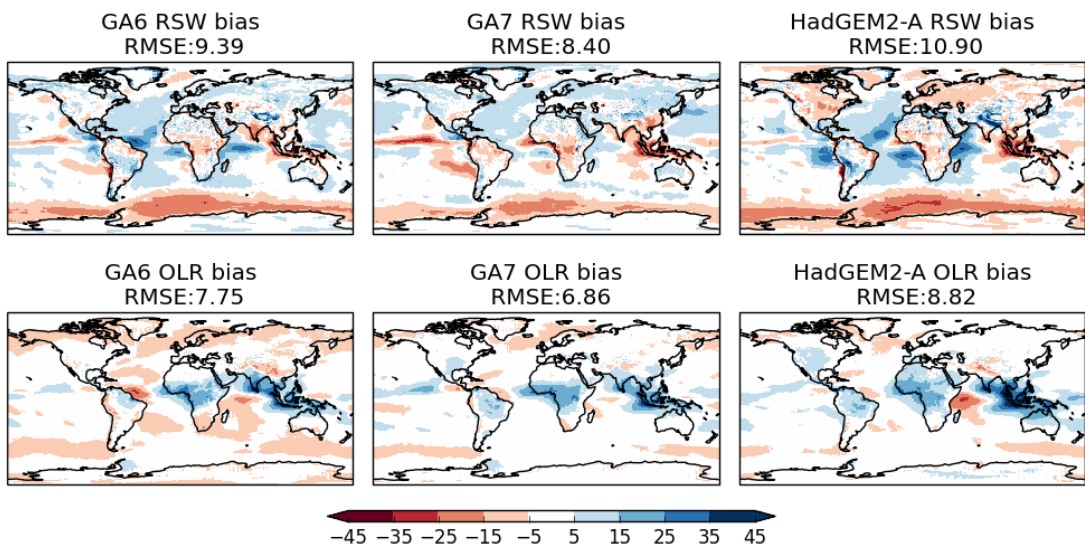

Figure 13: Multi-annual mean bias in RSW (top) and OLR (bottom) ($Wm^{-2}$) against CERES-EBAF for GA6, GA7 and HadGEM2-A. The spatial root-mean-square error (RMSE) is shown at the top of each panel.

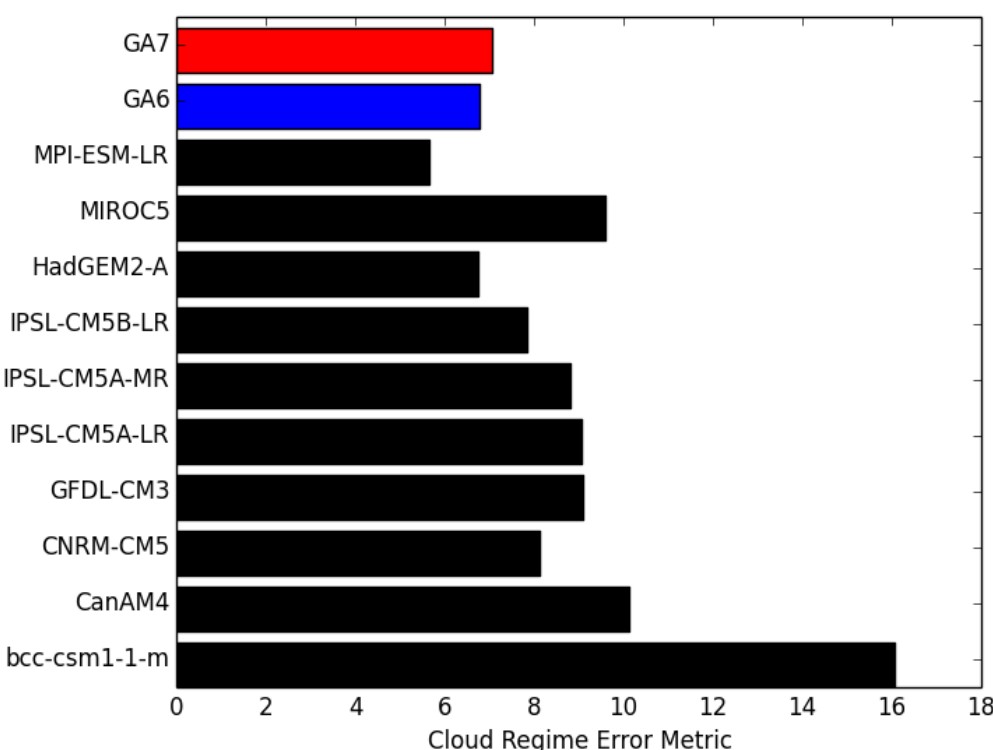

Figure 14: Cloud Regime Error Metric (CREM$_{pd}$) from Williams and Webb (2009) for the global cloud regimes of Tselioudis *et al* (2013) calculated for GA6 (blue), GA7 (red) and all of the CMIP5 models which have the required diagnostics available (black). Zero represents a perfect score with respect to the ISCCP observations.