# Peer review of "A multi-diagnostic approach to cloud evaluation"

_Geoscientific Model Development, 2016_

## Referee Comment (RC2) · Anonymous Referee #2 · 7 Mar 2017

General comments:

This paper illustrates a model evaluation against a variety of different observational datasets of cloud and radiative properties. A variety of approaches are used to limit analysis to specific cloud regimes and to specific time-scales.

The paper mainly serves as an example of how model evaluation against available satellite and ground-based observations of cloud properties might be performed, including the use of techniques to account for uncertainties or biases in satellite retrievals (using simulators), techniques to isolate specific cloud or dynamical regimes (using compositing), and techniques to isolate the climatological or systematic biases in the model from short-timescale processes (using hindcasts). While this is a useful contribution, the paper leaves much to be desired in terms of physical interpretation,

attribution, and discussion of identified biases, and instead focuses primarily on listing the identified biases.

Additionally, there is little (if any) discussion of uncertainties in the observational products used, or of the uncertainties in the comparisons between the model fields and those observations.

In light of these shortcomings, I would recommend major revisions to the paper, in particular to dive somewhat deeper into identifying physical processes responsible for the identified biases in the model in terms of the model formulation.

Specific comments:

140: A definition of low, mid, and high cloud categories should be provided here (i.e., what are the altitude bounds for each category?). A short description of how these histograms are produced would also be useful to the reader here, in addition to providing the reference provided (i.e., cloud occurrence in each category is defined as that which exceeds a minimum backscatter ratio of ??).

153: A brief explanation of the approach for each simulator would be helpful here (i.e., the ISCCP simulator emulates the way the retrieval infers cloud top pressure by estimating brightness temperature...).

156-165: The addition of this diagnostic the combines the CALIPSO and CloudSat hydrometeor occurrence is fantastic, but this description and discussion of the implementation is not nearly sufficient. A much more thorough description of the algorithm should be provided. The rationale for the choice of thresholds used seems somewhat incomplete as well, and it would be nice to see the comparison between GOCCP and RL-GEOPROF referred to on line 159. On line 160 it is suggested that the cloud detection algorithms differ between that used in COSP and that in RL-GEOPROF, but the nature of this difference is not explicitly stated and probably should be. Overall, some discussion of the uncertainties and sensitivities to the formulation of this new diagnostic should probably be provided to justify its use in the model evaluation. This could potentially be a significant contribution of this paper.

212-215: This is a nice result, and it would be worth expanding on the cause for the difference in cirrus between GA6 and GA7. In particular, some justification for the claim that the largest difference is due to the reduction in the rate of cirrus spreading could be shown, such as a figure showing the cirrus amount in GA7 with and without the adjusted cirrus spreading parameterization. I do not think the formulation of the cirrus spreading parameterization, or the changes made to improve the simulation, have been documented well enough in the manuscript. This result showing the decrease in cirrus and better agreement with both CALIPSO and CloudSat is a nice validation of the improvement in the simulation due to these changes, and would go nicely with a more thorough explanation of what is going on here.

221-222: How do we know that the revised numerics are responsible for the improvement in GA7? What specifically changed in the formulation of the model?

230: How is the "grid-box cloud fraction" being calculated? I am somewhat confused as to how this is produced alongside the profiles of reflectivity shown in the top panel. Is cloud fraction simply being aggregated onto a coarser grid from the reflectivity, calculated as the fraction within the coarser bins above some reflectivity threshold?

232-236: What does this imply about the model formulation (the cloud parameterizations)?

242: Add a note here that the drizzle rates cited are not shown here.

247-250: This is a nice demonstration of the impact of the new microphysics package, but this is lacking a discussion of the mechanisms for the improvement, and should be accompanied by a description of the changes.

258: Could the increase in cirrus here be explained by excessive advection of the cirrus outflow, or again maybe something to do with the cirrus spreading parameterization

referred to earlier? What is responsible for the improvement in GA7?

261-270: This discussion does not contain much substance, and inclusion of the ISCCP comparison seems to almost be an afterthought. This either needs a more complete treatment of the sources of differences, or consider cutting from the manuscript to make room for some of the more fleshed out analysis, such as the discussion of improvements in thin cirrus.

278-279: This statement could use evidence or a citation to back it up.

281-286: This could be better tied in with the discussion of cirrus above. In general though the results from this figure are not very compelling and do not seem to add much to the discussion. It is also not clear to me from Figure 5 that cirrus is overestimated in GA6. The most apparent biases in this figure are the altitude bias in the location of the cirrus maximum in GA6, and an overall underestimation of cirrus in GA7.

287-290: These conclusions are difficult to draw from Figure 5 as shown due to the scales of the axes used. If boundary layer cloud is the focus of this figure, it would be better to show just the boundary layer for the lower panel (SST composites), and on a cloud fraction scale that allows the reader to actually see the differences between the different curves.

340: I realize this is explained in the cited manuscript, but at least a simple explanation of the equation tested should be given here.

352: "Reasonably good" is awkward language to use here. I would suggest replacing with something like "while the cloud simulation was in reasonable agreement with observations".

356: Again "reasonably good" is awkward here.

368-369: Elaborate on how these biases are consistent with the radiation errors.

385-389: This is an excellent example of the utility of using multiple observations in the

evaluation strategy. This would be a good point to emphasize, and perhaps use as a jumping off point for a more elaborate investigation of the source of these differences (multi-layered cloud vs excess precipitation) than is given in the sentences to follow.

401: Why is SYNOP data the most reliable here?

403-405: Need evidence or references to back this up.

410: How is an okta defined in the context of the model?

439: What caused the reduction in the cold bias in GA7?

447-450: I am not sure I entirely agree with these conclusions. The reflected shortwave biases around the subtropical cumulus transitions seem to have reversed in sign between HadGEM2 and GA7, but the magnitudes do not seem to be universally reduced. Perhaps I am looking at the wrong part of the figure though, so maybe a box or symbol on the figure indicating the region where the improvement is evident would be appropriate. The underestimate in reflective shortwave over the Southern Ocean also does not appear to be significantly reduced.

482-485: This seems to really be a key point of the paper: to demonstrate that the multi-diagnostic approach used reduces the possibility of drawing the wrong conclusions. This is hinted to at points in the paper, but I think this could be drawn together a little better here, perhaps by recounting the points in the preceeding analysis that illustrate this (such as the contrast in the comparisons between CloudSat and CALIPSO that demonstrate errors due specifically to thin cirrus, or to excess precipitation as opposed to cloud errors).

---

## Author Comment (AC1) · 30 Mar 2017

Keith D. Williams and Alejandro Bodas-Salcedo

keith.williams@metoffice.gov.uk

**Author's response to referee 1 on "A multi-diagnostic approach to cloud evaluation"**

K. D. Williams and A. Bodas-Salcedo

March 30, 2017

**1 Major comments from referee 1**

**1.1 Referee Comments**

My primary concern is on the scientific focus of this study. The title of the paper seems to suggest that the aim of this work is to introduce "a multi-diagnostic approach to cloud evaluation". However, the paper has spent a lot of time on the inter-comparison of the two configurations of the UM model. I have no problem with whichever topic the study is designed to focus on, as both topics have their own values. However, since the study "tries" to cover two topics at a time, the discussions are somewhat lacking in depth. Therefore, the paper reads more like a report.

If the study is designed to focus on introducing a new multi-diagnostic approach, then a thorough introduction of this approach, including the developments of individual diagnostic methods (including necessary technical details), their merits and limitations, their applications in the literature, as well as a quantitative estimate of the uncertainties

of these methods, should be fully discussed. The authors have discussed some of the above mentioned aspects, but only to a very limited extent.

If the study is designed to focus on the evaluation of the simulations, then I have real trouble in understanding what have been done in the new configuration. Section 2a provides a general summary of the changes that have been made, but necessary details such as what processes or parameters have been added or changed in the parameterizations are lacking. Also, there is no dedicated case study to investigate the model performance in depth (except a snapshot in Figure 3 and Figure 8). As such, it is very difficult for a reader to appreciate what differences in the simulations can be considered as a real improvement. This is particularly true when considering the presence of new errors in the new configuration for some cloud properties.

**1.1.1 Author's response**

The aim of the paper is to show how a comprehensive approach to cloud evaluation can be valuable in developing and assessing a new model configuration. In order to achieve this aim we believe we need to show both the multi-diagnostic approach and how it can be used to assess the performance of a new model configuration against a control. However, in the revised manuscript we address the referee's concern that not enough depth of information is given. We have deliberately taken the approach of, wherever possible, using published methods, hence a technical discussion of these methods already exists in the published literature and we believe it would make the paper too cumbersome to repeat it all. The novel aspect here is that we draw the techniques together for the purpose of assessing a new model configuration as part of the model development process. Hence we have provided more detail of the parametrization changes made in the model development process and the relative merits of the different diagnostic approaches (e.g. why one observational dataset might be chosen over another to look at a particular aspect of the cloud simulation). We feel that this discussion of the diagnostic approach is best placed within the results sections to highlight the point that the chosen approach will vary depending on the particular characteristic being examined.

**1.1.2 Manuscript changes**

Section 2a has been considerably expanded with a more thorough description of the relevant parametrization changes. Within section 3, where possible, the text attributing changes in the errors to particular parametrization changes has been expanded (e.g. around the warm rain microphysics discussion) to discuss how the parametrization differences lead to the improvement and the physical processes operating. The results of two new simulations have been added to Figure 2 in order to clearly attribute the differences seen to particular parametrization changes.

The description of the observational datasets and, where relevant to the paper, their uncertainties has been expanded in Section 2b and in the results sections. In a number of places, we have enhanced the discussion of the value of the multi-diagnostic approach and the increased process-orientated understanding it can provide (e.g. around the mid-latitude cyclone RSW error and cloud errors over mid-latitude land).

**1.2 Referee Comments**

My second concern is on the comparison of model simulations against satellite observations (e.g. Figure 7, 9, 10, and similarly supplementary Figure 2-4). Many differences are discussed; however, these is no discussion on their statistical significance. How much of the difference is due to sample errors and how much is due to systematic errors in the model? In my view, a significance test should be applied to the analysis to insure that the differences discussed are meaningful. To do this you can use something simple such as a t-test or more appropriately a Monte Carlo method as applied

in Booth et al. (2013).

**1.2.1 Author's response**

We have now conducted a t-test based on the inter-annual variability of the observations and the models for the figures the referee refers to (and several others where this could be considered an issue). As expected, all the systematic errors discussed in the paper are considerably larger than the inter-annual variability and so remain significant.

**1.2.2 Manuscript changes**

Figures 2, 4, 5 and 11 have been reprocessed with shading around the line plots to represent 5% significance. For Figures 7, 9 & 10, the region of $<$5% significance has been coloured white so that all coloured regions in these plots show statistically significant differences. The significance test is also referred to in section 2b.

**2 Specific comments from referee 1**

**2.1 Comment**

Line 62-63: that's fine, but you also have spent a lot of time on inter-comparison of the two configurations of the model.

**2.1.1 Response & manuscript change**

The purpose of the paper is to show how a comprehensive approach to cloud evaluation can be valuable in developing and assessing a new model configuration. The

sentence has been altered in the revised manuscript to indicate this.

**2.2 Comment**

Line 66: "high", "mid", and "low" clouds need to be defined.

**2.2.1 Response & manuscript change**

Definitions have been added to the manuscript as low:>680hPa, mid:440hPa–680hPa, high:<440hPa.

**2.3 Comment**

Line 73: please define "NWP".

**2.3.1 Response & manuscript change**

This was already defined on line 24.

**2.4 Comment**

Line 97-117: a summary of the changes is good, but what changes have actually been made? What processes or parameters have been added or modified in the parameterizations? For example, what has been changed in the auto-conversion scheme (line 101)? What does the change do in the new aerosol scheme (line 112)? What does the turbulent scheme do to the production of liquid water (line 110)? You have provided the references, but some necessary details would be appreciated by the readers and

**GMDD**

would help justify your argument of the model improvement.

**2.4.1 Response & manuscript change**

This section of the paper is considerably expanded in the revised manuscript with a more detailed explanation of the parametrization changes as the referee requests. It should also be noted that it is intended that the present paper will be included within a GMD special issue which will also include the GA7 model description paper containing a full documentation of all the parametrization changes.

**2.5 Comment**

Line 145 and 147: CloudSat and CALIPSO provide a "curtain view" of the clouds, which are not really 3-D.

**2.5.1 Response & manuscript change**

'3D structure' has been replaced with 'hydrometeor profile'

**2.6 Comment**

Line 180: so how many years are used exactly?

**2.6.1 Response & manuscript change**

The following has been added to the revised manuscript "25 years for ISCCP, 12 years for CERES-EBAF and 5 years for CloudSat/CALIPSO"

**2.7 Comment**

Line 194: but you said "3-D" before

**2.7.1 Response & manuscript change**

The '3D' on line 146 (now removed) referred to CloudSat. Here we are stating that CALIPSO provides the best 2D (latitude/longitude) estimate of total cloud fraction; this doesn't preclude it from having useful information in the vertical as well.

**2.8 Comment**

Line 212: "this corrected in GA7" should be "this is corrected in GA7".

**2.8.1 Response & manuscript change**

Revised manuscript has been corrected as reviewer suggests.

**2.9 Comment**

Line 219: it does appear to be the case in GA6 to me. Please clarify.

**2.9.1 Response & manuscript change**

Referring to the top left panel of Figure 2b we can see no evidence that the altitude of the cirrus with lower backscattering ratios (3-5) is any higher than the thicker cirrus (backscattering ratios 7-20) - if anything the reverse is true. This is unlike the panels for

GA7 and CALIPSO which show the cirrus with the lowest backscattering ratios to be higher. We really can't see how we can make this clearer and request that the referee looks again at the text and figure.

**2.10 Comment**

Line 224: in this case I see the model produces a lot of mid-top clouds (which seem to have moderate optical depth) whereas you argue earlier (line 191) that the model simulates too little of this type of cloud?

**2.10.1 Response & manuscript change**

The hydrometeor signal observed is likely to be the spurious large scale precipitation referred to in the discussion of Figure2c and result from thin large scale cloud which has formed in the moist air around the convective system. As they occur under the anvil of a deep-convective system they won't be seen by ISCCP, and most of them may not be seen by CALIPSO either due to full attenuation from the ice cloud above. In contrast, the mid-top cloud which is 'missing' should be visible to CALIPSO (almost certainly it is missing congestus-type cloud). An extra paragraph has been added to the manuscript discussing these points.

**2.11 Comment**

Line 230: how "cloud fraction" is defined in the simulation and in the observational data set, respectively? Is a direct comparison meaningful? Please clarify.

**2.11.1 Response & manuscript change**

The radar–lidar product has considerably higher along track resolution (nominally 1.7km) than the model (80km at the equator), hence regridding the combined radar-lidar data onto the model grid gives an observed cloud fraction to a precision of about 2%. The main assumption here is that the along-track cloud fraction is representative of the 2D grid box. Whilst this is a fair assumption when considering a large number of cases which the A-train will cross at random orientations, we acknowledge that there may be an error when considering a single case such as this. However it's unlikely to affect the key model errors discussed in the paper regarding the figure. These points and caveats have been expanded upon in the revised manuscript as the reviewer suggests.

**2.12 Comment**

Line 238-239: a lot of these "drizzling" clouds in the simulations have a reflectivity below -20 DBZ, which is very, very weak. It seems odd that these clouds are not picked up by the CALIPSO simulator at all.

**2.12.1 Response & manuscript change**

As we note in the paper, the rates are $<0.005mm/hr$ which is consistent with the very weak signal. The concrete evidence given in the paper is that if large scale rain is not passed to the CloudSat simulator then the signal is removed. As we are below a thick anvil, the CALIPSO simulator signal is likely to have been attenuated and so not see cloud if present. However in GA7, once the spurious precipitation is removed, there is still a cloud signal in the CloudSat simulator just below the threshold of -40dBZ which suggests that the cloud is very thin.

**2.13 Comment**

Line 257-258 and relevant texts throughout the paper: care should be taken when drawing this conclusion. Previous studies (e.g. Chepfer et al. 2013) have shown that, due to the averaging issue, differences in the zonal cloud fraction retrieved in different CALIPSO products can be quite large (up to a factor of 2 for some regions). It is not unlikely that the GOCCP may have underestimated the cirrus extent.

**2.13.1 Response & manuscript change**

The reviewer is correct, and GOCCP probably underestimates the amount of cirrus. Chepfer at al. (2013) show that the averaging effect is sensitive to the length of the averaging and is higher for low-level, small-scale broken cloud. For high clouds, the differences between GOCCP and the CALIPSO cloud retrieval used by RL-GEOPROF are dominated by the SR detection threshold. The height-dependent SR detection threshold used in this study increases the sensitivity to high clouds (supplementary Figure 1). For cirrus clouds in the regions shown in Figure 2, the bias introduced by lack of averaging smaller than 0.05 (Figure 10 in Chepfer at al., 2013). This supports the interpretation that GA6, and to a lesser extent GA7, overestimates cirrus. This discussion has been added to the manuscript.

**2.14 Comment**

Line 298: this is a fairly big box. While I understand that this is a standard method used in previous studies, I am not convinced that it is appropriate for high-latitude regions, where cyclones (e.g. polar vortex) are generally smaller in size and the distance between individual cyclones can be a lot smaller (compared to mid-latitude cyclones).

**2.14.1 Response & manuscript change**

Throughout the paper we have tried to use published methodologies. As the referee notes, this is a standard size used in other studies looking at similar latitude bands. We acknowledge that cyclones come in a range of sizes and that this technique will just combine them all and hence smooth the signal (this point has been added to the revised manuscript), however we have tested different box sizes and the results are qualitatively similar and conclusions unchanged.

**2.15 Comment**

Line 331-322: This is a complicated case, with multiple fronts being diagnosed. Therefore it is hard for me to associate the cloud features discussed in the paper to the synoptic components shown on the MSLP chart. Further information such as latitude and longitude on the discussed cloud fields should help.

**2.15.1 Response & manuscript change**

Latitude references have been added to the revised manuscript as the reviewer suggests.

**2.16 Comment**

Line 332-334: I don't understand this sentence.

**2.16.1 Response & manuscript change**

This and adjacent sentences have been re-written to explain the point more clearly and a reference added as the result is consistent with previous experience.

**2.17 Comment**

Line 364 and relevant text later in the paper: I disagree. What I have seen is that the large RSW bias is present in some of the cold air side of the cyclone, but almost everywhere in the poleward side of the cyclone! Why? This is, to me, an important issue but no discussion has been made in the paper (or the referenced study). There is a lot of focus on the cold air side of the cyclone, but this is only part of the story revealed by the plot. Also, the bias on the cold air side of the cyclone does not explain the poleward increase of the radiative bias shown in Figure 13.

**2.17.1 Response & manuscript change**

We don't really understand the referee's distinction between the "cold air" and "poleward" side of the cyclone. On average, the poleward side of the cyclone is a subset of the cold air which also extends around the western side of the cyclone. By "cold air" we are referring to all of the region from the poleward side of the cloud head associated with the typical warm front position of about 3 o'clock on the southern hemisphere composite in figure 9, around the poleward and westward side to the poleward edge of the typical cold front position at 10-11 o'clock. In the revised manuscript we have clarified that cold air includes the poleward side of the cyclone.

**2.18 Comment**

Line 368: I don't think Figure 10 is necessary. It does not seem to provide any substantially different information than Figure 9.

**2.18.1 Response & manuscript change**

The key difference is that RSW in Figure 9 will depend on the insolation, hence for the same cloud albedo error, the RSW error will be larger in the summer than winter. Figure 10 shows the in-cloud albedos which do not depend on the insolation (this point has been explicitly added to the revised manuscript). Hence Figure 10 shows the interesting point that the in-cloud albedos are lower in the austral summer compared with the winter which, combined with the stronger insolation, leads to the larger negative RSW bias in Figure 9.

**2.19 Comment**

Line 369-370: why the cloud amount errors are not large enough to contribute significantly to the SW errors? Please explicate.

**2.19.1 Response & manuscript change**

We accept that this statement was probably too strong and requires correction which we have done in the revised manuscript. The cloud fraction errors for GA7 in Figure 7 and the supplementary material do not have the same spatial pattern (and are sometimes of the wrong sign) to explain the RSW error.

**2.20 Comment**

Line 373-374: again, the errors seem to be prevalent in the poleward side of the cyclone, too (which is also the case in the referenced study).

**2.20.1 Response & manuscript change**

See above response regarding "poleward" and "cold air".

**2.21 Comment**

Line 411: why this appears to be an issue for the UM? Please explicate.

**2.21.1 Response & manuscript change**

Sentence has been revised in the manuscript to "This appears to be an issue for the UM in parts of the tropics where too little shallow cumulus is simulated and typically the model has cloud fractions of <3 oktas whereas fractions over this threshold are often observed and hence a cloud base height assigned".

**2.22 Comment**

Line425-426: could some of these excess low clouds actually be precipitation not being detected by the instrument?

[Figure]

**2.22.1 Response & manuscript change**

Here we are comparing the model cloud (no simulator involved) with SYNOP observations so the presence, or not, of precipitation should be irrelevant.

**2.23 Comment**

Line 447-448: but now there seems to be too much red (for RSW) in the sub-tropics which was non-existent in HadGEM2-A?

**2.23.1 Response & manuscript change**

The sentence has be revised to read "The error in the sub-tropical cumulus transition regions of excess RSW has been removed and there is now a smaller negative bias in GA7". We have also reproduced Figure 13 with a revised colour bar to make it clearer that the negative bias in GA7 is smaller in magnitude than the positive bias in HadGEM2-A.

**2.24 Comment**

Figure 2: (1) you use "equivalent reflectivity factor" in the plot but "reflectivity" in the caption (and the following figures). (2) What do the colour bars show? It should be indicated in the plots. (3) The lowest km should be masked in the CloudSat plot (as done in your following figures).

**2.24.1 Response & manuscript change**

The term "reflectivity" is now used throughout (including figures). Indication of what the colour bars show has been added to the Figure 2 caption and explanation of how the histograms are formed has been added to section 2. The lowest 1.2km has now been masked for CloudSat (the main instrument to suffer from near-surface issues).

**2.25 Comment**

Figure 3: What does "CloudSat/CALIPSO" mean in the top left plot while you only show reflectivity?

**2.25.1 Response & manuscript change**

The caption to the figure explains that "...situations in which the lidar detected cloud but the radar did not being included with a nominal value of -40dBZ (e.g. Mace and Wrenn, 2013)"

**2.26 Comment**

Figure 5: I don't think including the very cold SST ranges is necessary as they are quite rare and the plots show very similar features (i.e. the first four plots in the bottom panel).

[Figure]

**2.26.1 Response & manuscript change**

Although relatively rare spatio-temporally, these typically represent the subtropical stratocumulus regions which are widely regarded by the cloud feedback community to be critical for the cloud response to climate change, hence their separation by SST is relevant. In addition, the coldest SST bin shows a significant (i.e. still distinguishable following the significance tests discussed above have been applied) error in GA6 which has been improved in GA7.

**2.27 Comment**

Figure 8: is 64.32N (the right end of the cross-session plots) over land? I can see the topography-like feature at the surface in the bottom left plot, but why there are clouds produced underneath the surface in the simulations?

**2.27.1 Response & manuscript change**

Neither cross section is over land. The masked region in the lower-right corner corner of the observed cross-section is due to the reliability mask associated with the RL-GEOPROF product indicating high uncertainty in both CloudSat and CALIPSO for these bins.

---

## Author Comment (AC2) · 30 Mar 2017

11

**Author's response to referee 2 on "A multi-diagnostic approach to cloud evaluation"**

K. D. Williams and A. Bodas-Salcedo

March 30, 2017

**1   Major comments from referee 2**

**1.1   Referee Comments**

The paper mainly serves as an example of how model evaluation against available satellite and ground-based observations of cloud properties might be performed, including the use of techniques to account for uncertainties or biases in satellite retrievals (using simulators), techniques to isolate specific cloud or dynamical regimes (using compositing), and techniques to isolate the climatological or systematic biases in the model from short-timescale processes (using hindcasts). While this is a useful contribution, the paper leaves much to be desired in terms of physical interpretation, attribution, and discussion of identified biases, and instead focuses primarily on listing the identified biases.

Additionally, there is little (if any) discussion of uncertainties in the observational products used, or of the uncertainties in the comparisons between the model fields and

those observations. In light of these shortcomings, I would recommend major revisions to the paper, in particular to dive somewhat deeper into identifying physical processes responsible for the identified biases in the model in terms of the model formulation

**1.1.1 Author's response**

We have included further detail regarding changes made to the parametrizations between GA6 and GA7 and have provided further evidence for attributing identified changes in cloud errors to particular model improvements. It should also be noted that it is intended that the present paper will be part of a GMD special issue which will also include the GA7 model description paper, so a complete description of the model will be readily available. We wish to avoid including speculation in the paper, however to the extent possible, in the revised manuscript we have attempted to link the errors to known model issues.

Observational uncertainties are complex. They depend on the details of the scene being observed (e.g. cloud size, height), illumination conditions, etc. Therefore, a full description of observational uncertainty is not possible within the scope of this paper. We have opted for bringing in information on observational uncertainty when appropriate within the discussion of the results (see response to specific comments).

**1.1.2 Manuscript changes**

Section 2a has been considerably expanded with a more thorough description of the relevant parametrization changes. Within section 3, where possible, the text attributing changes in errors to particular parametrization changes has been expanded (e.g. around the warm rain microphysics discussion) to discuss how the parametrization differences lead to the improvement and the physical processes operating (see answers to specific comments). We have also added text to draw together the results from

different diagnostic techniques to provide greater process understanding of the errors (e.g. around the mid-latitude cyclone RSW error).

The results of two new simulations have been added to Figure 2 in order to clearly attribute the differences seen to particular parametrization changes.

In the revised manuscript, we provide greater discussion of the uncertainties in the observational products where they are relevant to the paper (e.g. the differences between GOCCP and the CALIPSO cloud retrieval used by RL-GEOPROF). The revised manuscript also includes estimates of significance associated with sampling error to the figures.

**2 Specific comments from referee 2**

**2.1 Comment**

140: A definition of low, mid, and high cloud categories should be provided here (i.e., what are the altitude bounds for each category?). A short description of how these histograms are produced would also be useful to the reader here, in addition to providing the reference provided (i.e., cloud occurrence in each category is defined as that which exceeds a minimum backscatter ratio of ??).

**2.1.1 Response & manuscript change**

Definitions have been added to the manuscript as low:>680hPa, mid:440hPa–680hPa, high:<440hPa along with a description of the histogram as the referee suggests.

**2.2 Comment**

153: A brief explanation of the approach for each simulator would be helpful here (i.e., the ISCCP simulator emulates the way the retrieval infers cloud top pressure by estimating brightness temperature...).

**2.2.1 Response & manuscript change**

A brief description of each simulator has been added as the referee suggests.

**2.3 Comment**

156-165: The addition of this diagnostic the combines the CALIPSO and CloudSat hydrometeor occurrence is fantastic, but this description and discussion of the implementation is not nearly sufficient. A much more thorough description of the algorithm should be provided. The rationale for the choice of thresholds used seems somewhat incomplete as well, and it would be nice to see the comparison between GOCCP and RL-GEOPROF referred to on line 159. On line 160 it is suggested that the cloud detection algorithms differ between that used in COSP and that in RL-GEOPROF, but the nature of this difference is not explicitly stated and probably should be. Overall, some discussion of the uncertainties and sensitivities to the formulation of this new diagnostic should probably be provided to justify its use in the model evaluation. This could potentially be a significant contribution of this paper.

**2.3.1 Response & manuscript change**

As the referee requests, this paragraph has been completely re-written and expanded to provide a more detailed description of the diagnostic along with justification of the

choices made.

**2.4 Comment**

212-215: This is a nice result, and it would be worth expanding on the cause for the difference in cirrus between GA6 and GA7. In particular, some justification for the claim that the largest difference is due to the reduction in the rate of cirrus spreading could be shown, such as a figure showing the cirrus amount in GA7 with and without the adjusted cirrus spreading parameterization. I do not think the formulation of the cirrus spreading parameterization, or the changes made to improve the simulation, have been documented well enough in the manuscript. This result showing the decrease in cirrus and better agreement with both CALIPSO and CloudSat is a nice validation of the improvement in the simulation due to these changes, and would go nicely with a more thorough explanation of what is going on here.

**2.4.1 Response & manuscript change**

Figure 2b now has two additional simulations added to it, one of which is GA6 but with the cirrus spreading rate reduced to the GA7 value to demonstrate the impact as the referee suggests. Discussion of this is expended where the figure is referred to in section 3 and the description of the cirrus spreading change in section 2a has been expanded to include the origin of the parametrization, how it is working and the justification for reducing this parameter.

**2.5 Comment**

221-222: How do we know that the revised numerics are responsible for the improvement in GA7? What specifically changed in the formulation of the model?

**2.5.1 Response & manuscript change**

Figure 2b now has two additional simulations added to it, one of which is GA6 but using the 6A convection scheme (revised numerics) to demonstrate that this is responsible for the increase in altitude of the cirrus. The description of the 6A convection scheme has been considerably expanded in section 2a with a list of the changes made to the formulation, however the increase is cirrus height is very much an outcome - it's not clear why these changes have this effect (other than the numerics are more accurate).

**2.6 Comment**

230: How is the "grid-box cloud fraction" being calculated? I am somewhat confused as to how this is produced alongside the profiles of reflectivity shown in the top panel. Is cloud fraction simply being aggregated onto a coarser grid from the reflectivity, calculated as the fraction within the coarser bins above some reflectivity threshold?

**2.6.1 Response & manuscript change**

Yes, the combined radar–lidar product has considerably higher along track resolution (nominally 1.7km) than the model (80km at the equator), hence regridding the combined radar-lidar data onto the model grid gives an observed cloud fraction to a precision of about 2%. This has been made clear in the revised manuscript.

**2.7 Comment**

232-236: What does this imply about the model formulation (the cloud parameterizations)?

**2.7.1 Response & manuscript change**

The following has been added to the manuscript "This is likely due to too little condensate being detrained at these altitudes, with what there is being either the result of convection going slightly deeper on occasional timesteps or, more likely, some of the condensate being advected vertically having been detrained below."

**2.8 Comment**

242: Add a note here that the drizzle rates cited are not shown here.

**2.8.1 Response & manuscript change**

Added in the revised manuscript as the referee suggests.

**2.9 Comment**

247-250: This is a nice demonstration of the impact of the new microphysics package, but this is lacking a discussion of the mechanisms for the improvement, and should be accompanied by a description of the changes.

**2.9.1 Response & manuscript change**

The description of the warm rain microphysics scheme has been expanded in section 2a. We have also added the following in section 3 where the attribution of the change to the warm rain microphysics package is discussed "Within this package, the change to use the Khairoutdinov and Kogan (2000) scheme reduces auto-conversion rates by

a factor of around 100 compared with the scheme used in GA6. These rates would be too low without the Boutle et al. (2014) GCM upscaling, however even after this correction, the auto-conversion rates remain around 10 times small than GA6 which accounts for the removal of the spurious drizzle."

**2.10 Comment**

258: Could the increase in cirrus here be explained by excessive advection of the cirrus outflow, or again maybe something to do with the cirrus spreading parameterization referred to earlier? What is responsible for the improvement in GA7?

**2.10.1 Response & manuscript change**

The improvement in GA7 is due to the cirrus spreading change and this has now been added to the manuscript. The upper tropospheric wind errors are not large enough for the bias to be attributable to excessive advection, hence we retain the suggestion in the text as "possibly due to errors in microphysical processes, or macrophysical fields (such as relative humidity being too high)."

**2.11 Comment**

261-270: This discussion does not contain much substance, and inclusion of the IS-CCP comparison seems to almost be an afterthought. This either needs a more complete treatment of the sources of differences, or consider cutting from the manuscript to make room for some of the more fleshed out analysis, such as the discussion of improvements in thin cirrus.

**2.11.1 Response & manuscript change**

As we describe in the text, accurate simulation of cloud in this region is believed to be particularly important in determining the global cloud feedback under climate change. For this reason, the excellent simulation of stratocumulus amount is worth showing, however it hasn't changed much between the two configurations shown, hence the brevity of the paragraph. We have expanded the discussion around the ISCCP comparison since it highlights one of the key outstanding errors which remain, namely that in many regions low cloud remains too reflective. We have added "Consistent with this, comparison against a number of observational datasets indicates that the cloud effective radius simulated by the model is too low in many regions, including in subtropical stratocumulus, and is indicative of the aerosol cloud indirect effect being too strong."

**2.12 Comment**

278-279: This statement could use evidence or a citation to back it up.

**2.12.1 Response & manuscript change**

This was based on personal experience. Whilst we believe it correct, we do not have a reference and have therefore removed the statement from the revised manuscript.

**2.13 Comment**

281-286: This could be better tied in with the discussion of cirrus above. In general though the results from this figure are not very compelling and do not seem to add much to the discussion. It is also not clear to me from Figure 5 that cirrus is overestimated in GA6. The most apparent biases in this figure are the altitude bias in the location of the

cirrus maximum in GA6, and an overall underestimation of cirrus in GA7.

**2.13.1 Response & manuscript change**

The manuscript has been re-worded to link back to the discussion of the tropics as a whole. We now highlight the cirrus height increase and refer to the cirrus amount as a change rather than a universal improvement. This variance in whether the change is an improvement or detriment across the regimes highlights the importance of this figure in providing information over what was in the tropical mean analysis in Figure 2 - a point which has been added to the manuscript.

**2.14 Comment**

287-290: These conclusions are difficult to draw from Figure 5 as shown due to the scales of the axes used. If boundary layer cloud is the focus of this figure, it would be better to show just the boundary layer for the lower panel (SST composites), and on a cloud fraction scale that allows the reader to actually see the differences between the different curves.

**2.14.1 Response & manuscript change**

As the referee suggests, the lower panel of figure 5 has been re-drawn to just show the lowest few km and the cloud fraction scale adjusted to make it easier to view the differences.

**2.15 Comment**

340: I realize this is explained in the cited manuscript, but at least a simple explanation of the equation tested should be given here.

**2.15.1 Response & manuscript change**

The present manuscript has been revised to indicate that the change made to the equation when testing for anticyclones is identifying a local maxima in surface pressure rather than a local minima.

**2.16 Comment**

352: "Reasonably good" is awkward language to use here. I would suggest replacing with something like "while the cloud simulation was in reasonable agreement with observations".

**2.16.1 Response & manuscript change**

Sentence changed in the revised manuscript as reviewer suggests.

**2.17 Comment**

356: Again "reasonably good" is awkward here.

**2.17.1 Response & manuscript change**

Sentence changed to "Despite the cloud amount composites showing cloud fraction errors of less than 0.15 (and often less than 0.05) in GA7...".

**2.18 Comment**

368-369: Elaborate on how these biases are consistent with the radiation errors.

**2.18.1 Response & manuscript change**

Description expanded in revised manuscript to highlight that in regions of positive albedo bias in Figure 10, there is a positive RSW bias in Figure 9 and vice-versa. However, the in-cloud albedos in Figure 10 do not depend on the insolation hence for the same cloud albedo error, the RSW error will be larger in the summer than winter.

**2.19 Comment**

385-389: This is an excellent example of the utility of using multiple observations in the evaluation strategy. This would be a good point to emphasize, and perhaps use as a jumping off point for a more elaborate investigation of the source of these differences (multi-layered cloud vs excess precipitation) than is given in the sentences to follow.

**2.19.1 Response & manuscript change**

We have highlighted that this is a good example of the utility of using multiple instruments. We have also expanded the discussion explaining why we can't rule out either

the shielded low cloud or precipitation options at this stage (our suspicion is that both may contribute).

**2.20 Comment**

401: Why is SYNOP data the most reliable here?

**2.20.1 Response & manuscript change**

Sentence expanded in the revised manuscript to discuss the problems of viewing the lowest levels from space and that an upward pointing ceilometer or human observer is likely to be at their most accurate for low cloud bases.

**2.21 Comment**

403-405: Need evidence or references to back this up.

**2.21.1 Response & manuscript change**

Reference to Mittermaier (2012) added.

**2.22 Comment**

410: How is an okta defined in the context of the model?

**2.22.1 Response & manuscript change**

This is simply a cloud fraction of 1/8 (0.125). This has been added to the revised manuscript.

**2.23 Comment**

439: What caused the reduction in the cold bias in GA7?

**2.23.1 Response & manuscript change**

This was mainly due to the introduction of the 6A convection scheme. This has been added to the manuscript, however it is beyond the scope of this paper to discuss these non-cloud related impacts of the model changes and instead a reference given to Walters et al. (2017) who discuss this further.

**2.24 Comment**

447-450: I am not sure I entirely agree with these conclusions. The reflected short-wave biases around the subtropical cumulus transitions seem to have reversed in sign between HadGEM2 and GA7, but the magnitudes do not seem to be universally reduced. Perhaps I am looking at the wrong part of the figure though, so maybe a box or symbol on the figure indicating the region where the improvement is evident would be appropriate. The underestimate in reflective shortwave over the Southern Ocean also does not appear to be significantly reduced.

**2.24.1 Response & manuscript change**

The sentence has be revised to read "The error in the sub-tropical cumulus transition regions of excess RSW has been removed and there is now a smaller negative bias in GA7. The lack of RSW over the Southern Ocean has been reduced by a third and...". We have also reproduced Figure 13 with a revised colour bar to make it easier to quantify the changes e.g. that the negative bias in the transition region in GA7 is smaller in magnitude compared with the positive bias in HadGEM2-A.

**2.25 Comment**

482-485: This seems to really be a key point of the paper: to demonstrate that the multi-diagnostic approach used reduces the possibility of drawing the wrong conclusions. This is hinted to at points in the paper, but I think this could be drawn together a little better here, perhaps by recounting the points in the preceding analysis that illustrate this (such as the contrast in the comparisons between CloudSat and CALIPSO that demonstrate errors due specifically to thin cirrus, or to excess precipitation as opposed to cloud errors).

**2.25.1 Response & manuscript change**

The discussion has been expanded here using a number of examples including the ones the referee suggests.

---

## Author Response (AR2)

**Reponse to topical editor**:

*1) Page 5, new text: unfortunately, during revision of the text, some new notions have entered the text without explanation. These are: a) "the convection diagnosis parcel", b) "the (Zhang and Klein, 2013) data", c) "inhomogeneously forced".*

**These sentences have now been re-written to avoid the use of the notations.**

*2) Page 10, several places: Here you introduce a "scattering ratio". I believe this is what usually is called "backscattering ratio". If so, please use the standard terminology. Otherwise, "scattering ratio" must be defined.*

**Corrected to "backscattering ratio" as the editor suggests.**

*3) Page 16, ll. 374/375: the averaging effect of what?*

**Two additional senstences have been included ahead of this sentence to explain the context as being the averaging length used in the observational datasets. It is also discussed in section 2b.**

*4) Page 17, l. 380: I don't understand what this means: "by lack of averaging smaller than 0.05". Please rewrite.*

**Sentence re-written to make this clearer, and it follows on from the averaging discussion earlier in the paragraph which has now been clarified.**

*5) Figure 2: Please check whether the 1d and 2d histograms are all normalised to unity. For instance, it seems that the integrals under the curves in 2a, bottom right, are different; it seems also that the 2d integral in 2c, top left, is much larger than in the other histograms in that panel.*

**The histograms in Figure 2a should sum to the total cloud fraction (which they do) rather than unity as the clear-sky fraction is not included. Figure 2b,c&d should sum to the cloud fraction in each height bin i.e. each height**

bin contains a frequency histogram. Technically, calling them joint height–backscattering ratio/reflectivity histograms in the caption was incorrect and this has been changed in the revised version.

*6) Figures 7 and 9: please add "statistically" before "significant and provide the significance level selected (0.05 or 0.01 or even smaller?).*

**Changed as editor suggests.**

*Grammar and spelling errors.*

**All now corrected.**

[revised manuscript text omitted]